# QUERY-EFFICIENT OFFLINE PREFERENCE-BASED REINFORCEMENT LEARNING VIA IN-DATASET EXPLORATION

## ABSTRACT

Preference-based reinforcement learning has shown great promise in various applications to avoid reward annotations and align better with human intentions. However, obtaining preference feedback can still be expensive or time-consuming, which forms a strong barrier for preference-based RL. In this paper, we propose a novel approach to improve the query efficiency of offline preference-based RL by introducing the concept of in-dataset exploration. In-dataset exploration consists of two key features: weighted trajectory queries and a principled pairwise exploration strategy that balances between pessimism over transitions and optimism over reward functions. We show that such a strategy leads to a provably efficient algorithm that judiciously selects queries to minimize the overall number of queries while ensuring a robust performance. We further design an empirical version of our algorithm that tailors the theoretical insights to practical settings. Experiments on various tasks demonstrate that our approach achieves strong performance with significantly fewer queries than state-of-the-art methods.

## 1 INTRODUCTION

Reinforcement Learning (RL) has emerged as a powerful approach for solving a wide variety of sequential decision-making problems, including classic games (Silver et al., 2016), video games (Mnih et al., 2015; Vinyals et al., 2019), robotics (Ahn et al., 2022), and plasma control (Degrave et al., 2022) with supervision from just reward signals. Nevertheless, in many real-world applications, specifying an accurate reward function can be incredibly challenging due to the requirements of extensive instrumentation and a balance between multiple objectives. Therefore, preference-based RL (PbRL; Akrour et al., 2012; Christiano et al., 2017) has shown great promise since making relative judgments comparison is easy to provide yet information-rich. By learning preferences from human feedback, recent work has demonstrated that the agent can learn novel behaviors (Christiano et al., 2017; Kim et al., 2023) and achieve better alignment (Ouyang et al., 2022).

Despite its promise, an online PbRL approach requires a coupling of human evaluation and interaction with environments, rendering the process tedious and time-consuming. On the other hand, Offline PbRL (Shin et al., 2023; Kim et al., 2023) aims to utilize a pre-collected dataset to reduce the exploration cost. While offline PbRL disentangles online interaction and human evaluation, it limits possible queries within a predetermined dataset. This highlights the imperative to design a principled way to extract as much information as possible by actively making queries with the existing reward-free dataset while minimizing the number of queries made for preference feedback.

Designing query-efficient offline PbRL algorithms faces two main challenges: balancing exploration with preference queries and exploitation with a given dataset and learning temporal credit assignments with trajectory-wise comparison. To cope with these two challenges, we propose a

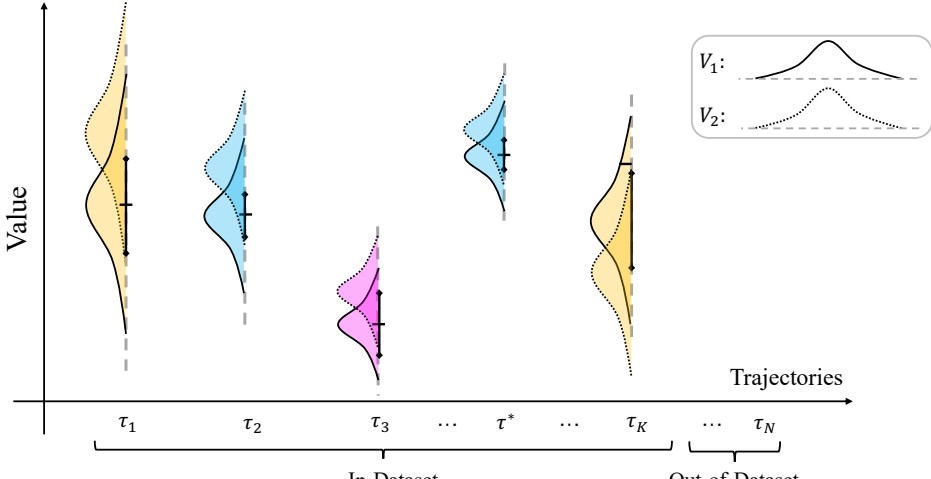

Figure 1: Illustration of principled pairwise exploration. The solid lines and dotted lines represent the estimation of two value function models $V_1$ and $V_2$, respectively. For efficient exploration, we first remove actions that are not optimal with high probability (i.e., $\tau_3$), then select two trajectory pairs such that the uncertainty of the preference between them is maximized (i.e., there is serious preference divergence between $\tau_1$ and $\tau_K$ since $V_2$ strongly prefer $\tau_1$ while $V_1$ strongly prefer $\tau_K$).

novel approach leveraging the concept of in-dataset exploration. For the first challenge, our method adopts a principled way of pairwise exploration, as depicted in Figure 1. For the second challenge, our method add temporal weights to the in-dataset trajectories to allow for fine-grained comparison, as depicted in Figure 2. We show that such a strategy is provably efficient for solving the problem, minimizing the overall number of queries while ensuring robust performance.

Building on the theoretical insight, we design an empirical version of our algorithm that extends the theoretical principles to practical settings. Experiments on navigation and manipulation tasks, including AntMaze (Fu et al., 2020) and Meta-World (Yu et al., 2019), show that our approach achieves strong performance with significantly fewer queries than state-of-the-art baselines.

In summary, our contributions are threefold: (1) we propose the concept of in-dataset exploration for offline PbRL; (2) we develop a provably efficient algorithm that uses a principled pairwise exploration strategy and achieves a balance between pessimism over transitions and optimism over rewards for in-dataset exploration; and (3) we design an empirical version of our algorithm and demonstrate its strong performance across various benchmarks and tasks.

## 1.1 RELATED WORK

**Preference-based RL.** Various methods have been proposed to utilize human preferences (Akrour et al., 2012; Christiano et al., 2017; Ibarz et al., 2018) and have been successful in complex control tasks (Christiano et al., 2017) and large language model alignment (Ouyang et al., 2022). For offline PbRL, OPRL (Shin et al., 2023) proposes a benchmark with several baseline methods. Kim et al. (2023) uses Transformer to model preference for better credit assignment. Kang et al. (2023) propose a direct method for preference-based policy learning. Theoretically, Pacchiano et al. (2021) proposes a provable PbRL algorithm in linear MDPs. Chen et al. (2022) generalizes it to cases where the Eluder dimension is finite. Zhan et al. (2023) studies offline PbRL with a given preference dataset.

**Reward-free Offline RL.** Yu et al. (2022) and Hu et al. (2023) utilize reward-free data to aid offline learning but assume a labeled offline dataset is available for learning the reward function. Ajay et al. (2020) utilizes reward-free datasets by extracting valuable behaviors. Ye et al.; Ghosh et al. (2023) consider the setting where both reward and action are absent in the dataset. We utilize offline reward-free data by allowing preference queries within the dataset.

## 2 PRELIMINARIES

### 2.1 LINEAR MDPS AND PERFORMANCE METRIC

We consider finite-horizon episodic Markov Decision Processes (MDPs), defined by the tuple $(\mathcal{S}, \mathcal{A}, H, \mathcal{P}, r)$, with state space $\mathcal{S}$, action space $\mathcal{A}$, horizon $H$, transition function $\mathcal{P} = \{\mathcal{P}_h\}_{h=1}^H, \mathcal{P}_h : \mathcal{S} \times \mathcal{A} \to \Delta(\mathcal{S})$, and reward function $r = \{r_h\}_{h=1}^H, r_h : \mathcal{S} \times \mathcal{A} \to [0, r_{\max}]$. For theoretical analysis, we consider the *linear MDP* (Yang & Wang, 2019; Jin et al., 2020) as follows, where the transition kernel and expected reward function are linear with respect to a known feature map.

**Definition 1** (Linear MDP). *An episodic MDP is a linear MDP with known feature map $\phi : \mathcal{S} \times \mathcal{A} \to \mathbb{R}^d$ if there exist unknown measures $\mu_h = (\mu_h^1, \ldots, \mu_h^d)$ and an unknown vector $\theta_h \in \mathbb{R}^d$ such that*

$$\mathcal{P}_h(s' \,|\, s, a) = \langle \phi(s, a), \mu_h(s') \rangle, \quad r_h(s, a) = \langle \phi(s, a), \theta_h \rangle \tag{1}$$

*for all $(s, a, s') \in \mathcal{S} \times \mathcal{A} \times \mathcal{S}$. And we assume $\|\phi(s, a)\|_2 \leq 1$ for all $(s, a, s') \in \mathcal{S} \times \mathcal{A} \times \mathcal{S}$ and $\max\{\|\mu_h(\mathcal{S})\|_2, \|\theta_h\|_2\} \leq \sqrt{d}$, where $\|\mu_h(\mathcal{S})\| := \int_{\mathcal{S}} \|\mu_h(s)\| ds$.*

A policy $\pi = \{\pi_h\}_{h=1}^H, \pi_h : \mathcal{S} \to \Delta(\mathcal{A})$ specifies a decision-making strategy in which the agent chooses actions adaptively based on the current state, i.e., $a_h \sim \pi_h(\cdot \,|\, s_h)$. The value function $V_h^\pi : \mathcal{S} \to \mathbb{R}$ and the action-value function (Q-function) $Q_h^\pi : \mathcal{S} \times \mathcal{A} \to \mathbb{R}$ are defined as

$$V_h^\pi(s) = \mathbb{E}_\pi \Big[ \sum_{t=h}^H r(s_t, a_t) \,\Big|\, s_h = s \Big], \quad Q_h^\pi(s, a) = \mathbb{E}_\pi \Big[ \sum_{t=h}^H r(s_t, a_t) \,\Big|\, s_h = s, a_h = a \Big], \tag{2}$$

where the expectation is w.r.t. the trajectory $\tau$ induced by $\pi$. We define the Bellman operator as

$$(\mathbb{B}_h f)(s, a) = \mathbb{E}_{s' \sim \mathcal{P}_h(\cdot|s,a)} \big[ r(s, a) + \gamma f(s') \big]. \tag{3}$$

We use $\pi^\star, Q_h^\star$, and $V_h^\star$ to denote the optimal policy, optimal Q-function, and optimal value function, respectively. We have the Bellman optimality equation

$$V^\star(s)_h = \max_{a \in \mathcal{A}} Q_h^\star(s, a), \quad Q_h^\star(s, a) = (\mathbb{B}_h V_{h+1}^\star)(s, a). \tag{4}$$

Meanwhile, the optimal policy $\pi^\star$ satisfies $\pi_h^\star(\cdot \,|\, s) = \mathrm{argmax}_\pi \langle Q_h^\star(s, \cdot), \pi(\cdot \,|\, s) \rangle_{\mathcal{A}}$.

We aim to learn a policy that maximizes the expected cumulative reward. Correspondingly, we define the performance metric as the sub-optimality compared with the optimal policy, i.e.,

$$\mathrm{SubOpt}(\pi, s) = V_1^{\pi^\star}(s) - V_1^\pi(s). \tag{5}$$

### 2.2 PREFERENCE-BASED REINFORCEMENT LEARNING

To learn reward functions from preference labels, we consider the Bradley-Terry pairwise preference model (Bradley & Terry, 1952) as used by most prior work(Christiano et al., 2017; Ibarz et al., 2018; Palan et al., 2019). Specifically, the probability of the preference label over two given trajectories $\tau_i \prec \tau_j$ is defined as

$$P\Big( \tau_i \prec \tau_j \,\Big|\, \theta \Big) = \frac{\exp \sum_{(s,a) \in \tau_j} r_\theta(s, a)}{\exp \sum_{(s,a) \in \tau_i} r_\theta(s, a) + \exp \sum_{(s,a) \in \tau_j} r_\theta(s, a)}.$$

The reward parameter can be learned with a given preference dataset $\mathcal{D}_{\text{pref}}$ with the following cross-entropy loss:

$$\mathcal{L}_{CE}(\theta) = - \mathop{\mathbb{E}}_{(\tau^1, \tau^2, o) \sim \mathcal{D}_{\text{pref}}} \left[ (1 - o) \log P_\theta(\tau^1 \succ \tau^2) + o \log P_\theta(\tau^2 \succ \tau^1) \right], \quad (6)$$

where $o$ is the ground truth label given by human labelers. In linear settings, the Bradley-Terry model is a special case of generalized linear preference models, defined as follows.

**Definition 2** (Generalized Linear Preference Models). *For the case of $d$-dimensional generalized linear models $P(\tau_1 \prec \tau_2) = \sigma(\langle \phi(\tau_1, \tau_2), \theta \rangle)$ where $\tau_i = (s_h^i, a_h^i)_{i=1}^H$ and $\sigma$ is an increasing Lipschitz continuous function, $\phi : \text{Traj} \times \text{Traj} \to \mathbb{R}^d$ is a known feature map satisfying $\|\phi(\tau_1, \tau_2)\|_2 \leq H$ and $\theta \in \mathbb{R}^d$ is an unknown parameter with $\|\theta\|_2 \leq S$.*

Throughout the paper, we consider the generalized linear preference model so that the Bradley-Terry preference model with linear MDP is a special case with $\sigma(x) = 1/(1 + e^{-x})$ and $\phi(\tau_1, \tau_2) = \sum_{h=1}^H (\phi(s_h^1, a_h^1) - \phi(s_h^2, a_h^2))$. Note that we overload the notation $\phi$ here for convenience. Our analysis can be readily generalized to general function approximation with finite Eluder dimensions (Russo & Van Roy, 2013; Chen et al., 2022).

## 3    IN-DATASET EXPLORATION

As mentioned in Section 1, there are two main challenges to making offline PbRL with in-dataset queries provably efficient: conducting proper exploration with a given dataset and learning proper temporal credit assignment with trajectory-based queries. The first challenge is due to the dilemma that we need to explore for preference learning while we also need to exploit for leveraging the offline dataset. The second challenge is due to the difficulty in learning temporal credit assignment with only trajectory-wise information. It is known to require an exponentially large amount of trajectories to learn a proper temporal credit assignment in the offline setting (Zhan et al., 2023).

For the first challenge, we propose a principled way for exploration that (1) makes a balance between pessimism over transitions and optimism for preference learning and (2) conducts pairwise exploration by first removing non-optimal trajectories and then selecting the most informative pair by maximizing value differences, as depicted in Figure 1. For the second challenge, we propose to add temporal weights on the fixed trajectories for fine-grained comparison. This helps us infer the preference on out-of-dataset trajectories, and an illustrative example is shown in Figure 2 (a) (b) (c). Performance comparison on the didactic chain MDP in Figure 2 (d) depicts the advantage of weighted queries on statistical efficiency. Please refer to Appendix E for more details on the chain MDP experiment.

Based on the above insights, we design an empirical algorithm, **O**ffline **P**reference-based **R**einforcement learning with **I**n-**D**ataset **E**xploration (OPRIDE) that augments trajectories in the dataset with random temporal weights and incorporates principled exploration strategies. OPRIDE solves the two main challenges above and achieves high query efficiency. A formal description of our proposed algorithm is provided in Algorithm 1.

### 3.1    THEORETICAL ANALYSIS FOR PRINCIPLED EXPLORATION

We first consider the setting where we are allowed to make online queries with a reward-free offline dataset. For a principled exploration strategy under such a setting, we can combine the wisdom from online PbRL and pessimistic value estimation for offline value estimation. Specifically, the strategy consists of two parts: (1) *pessimistic value iteration* (PEVI; Jin et al., 2021) as the backbone algorithm to account for the finite sample bias over dynamics due to a fixed offline dataset; (2) optimistic over preference functions for efficient exploration. To achieve such optimism, we first compute the

near-optimal policy for each query with the help of a confidence set over reward functions. Then, we select explorative policies that maximize the value difference within the policy set. Finally, we query the preference between the trajectories generated by the explorative policies and add it to the preference dataset. A detailed description is available in Algorithm 2.

**Pessimistic Value Estimation.** We use a pessimistic value function to account for offline error over dynamics due to finite data. Specifically, we consider *pessimistic value iteration* (PEVI; Jin et al., 2021), which adds a negative bonus over the standard $Q$-value estimation to account for uncertainty due to finite data. The negative bonus is propagated via the Bellman equation so that the output value function considers the uncertainty of the current step and that of future steps. Please refer to Algorithm 3 and Appendix B for more details.

**Construct Reward Confidence Set.** To estimate the reward parameter, we can use the cross entropy loss as in Equation 6 to get the MLE estimator $\widehat{\theta}_{\text{MLE}}$. However, the standard MLE estimator can have an unbounded norm, which motivates us to use a projected MLE estimator Instead, we consider the projected MLE estimator $\widehat{\theta}_k$ (Faury et al., 2020). Please refer to Appendix A.1 for more details on the projected MLE estimator. Then, the confidence set is given by

$$\mathcal{C}_k(\delta) = \left\{ \theta \;\middle|\; \text{s.t. } \|\theta - \widehat{\theta}_k\|_{\Sigma_k} \leq \beta_k(\delta) \right\} \tag{7}$$

where $\Sigma_k = \lambda I + \sum_{i=1}^{k-1} \left( \phi(\tau_i^1) - \phi(\tau_i^2) \right) \left( \phi(\tau_i^1) - \phi(\tau_i^2) \right)^\top$ with $\phi(\tau) = \sum_{h=1}^H \phi(s_h^\tau, a_h^\tau)$ and $\beta_k(\delta)$ is the confidence parameter.

It can be shown that with probability $1 - \delta$, the true reward function $\theta^\star$ is contained in the confidence set $\mathcal{C}_k(\delta)$ for all $k \geq 1$. Please refer to Appendix B for more details.

**Near-optimal Policy Set.** With the estimated reward model its confidence, we can define near-optimal policy set $\Pi_k$ as follows:

$$\Pi_k = \left\{ \widehat{\pi} \,\middle|\, \exists\, \theta \in \mathcal{C}_k(\delta), \widehat{\pi} = \text{greedy}\left( \widehat{V}_\theta \right) \right\}, \tag{8}$$

where $\widehat{V}_\theta$ is the output of Algorithm 3 with reward function $r_\theta$. Intuitively speaking, $\Pi_k$ consists of policies that are possibly optimal within the current level of uncertainty over reward and dynamics. By constraining policies in $\Pi_k$, We achieve proper exploitation by avoiding unnecessary explorations.

**Exploratory Policies.** For online exploration, we can choose two policies in $\Pi_k$ that maximize the uncertainty, and thus encourage exploration:

$$(\pi_h^{k,1}, \pi_h^{k,2}) = \underset{\pi_1, \pi_2 \in \Pi_k}{\text{argmax}} \; \underset{\theta_1, \theta_2 \in \mathcal{C}_k(\delta)}{\text{argmax}} \left( \left( \widehat{V}_{\theta_1}^{\pi_1} - \widehat{V}_{\theta_2}^{\pi_1} \right) - \left( \widehat{V}_{\theta_1}^{\pi_2} - \widehat{V}_{\theta_2}^{\pi_2} \right) \right) \tag{9}$$

Intuitively, we choose two policies $\pi_1, \pi_2$ to maximize the value difference, such that there is a $\theta_1 \in \mathcal{C}_k(\delta)$ that strongly prefers $\pi_1$ over $\pi_2$, and there is a $\theta_2 \in \mathcal{C}_k(\delta)$ that strongly prefer $\pi_2$ over $\pi_1$. Then we sample two trajectories $\tau^{k,1} \sim \pi^{k,1}, \tau^{k,2} \sim \pi^{k,2}$, query the preference between them, and add it to the preference dataset.

**Theoretical Guarantees.** We have the following theorem for our proposed Algorithm 2.

**Theorem 3.** *Suppose the underlying MDP is a linear MDP with dimension $d$ and horizon $H$. Then, with high probability, the suboptimality of Algorithm 2 is upper bounded by*

$$SubOpt(K) \leq \widetilde{\mathcal{O}}\left( \sqrt{\frac{C^\dagger d^2 H^5}{N}} + \sqrt{\frac{d^2 H^4}{K}} \right), \tag{10}$$

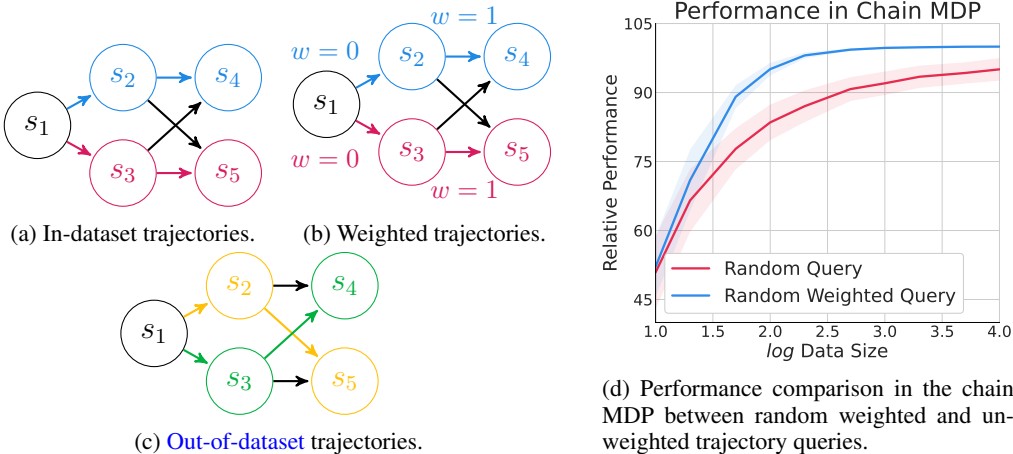

(a) In-dataset trajectories.

(b) Weighted trajectories.

(c) Out-of-dataset trajectories.

(d) Performance comparison in the chain MDP between random weighted and unweighted trajectory queries.

Figure 2: Illustration for the benefit of weighted trajectory queries. Suppose rewards are defined on states. Colored trajectories in (a) are contained in the offline dataset. Solely based on unweighted queries, we can not infer whether $s_2$ is better than $s_3$ and whether $s_4$ is better than $s_5$. Using additional weighted queries in (b) help us learn better credit assignment. In particular, weighted queries make the colored out-of-dataset trajectories in (c) distinguishable. In (d), the empirical result in the didactic chain MDP shows that the sample efficiency of random weighted queries outperforms random unweighted queries by a large margin.

*where $C^\dagger$ is the coverage coefficient in Definition 5, $N$ is the size of the offline dataset and $K$ is the number of queries.*

*Proof.* See Appendix C for a detailed proof. ☐

Equation 10 decomposes the suboptimality of Algorithm 2 into two terms nicely: the dynamics error term and the reward error term. The dynamics error is due to the finite sample bias of the dataset, and the reward error is due to the limited amount of preference queries. Compared to the dynamic error, the reward error is reduced by a factor of $\sqrt{H}$. Therefore, querying with an offline dataset can be much more sample efficient when $N \gg K$. This is due to the fact that the offline dataset contains rich information about dynamics and reduces the problem to horizonless bandits given infinite offline data (Hu et al., 2023). This also aligns with our empirical findings that $\sim 10$ queries are usually sufficient for reasonable performance in offline settings.

### 3.2 EMPIRICAL IMPLEMENTATION WITH WEIGHTED QUERIES

While Algorithm 2 has nice theoretical guarantees, it can be computationally inefficient to implement. Also, it assumes that we can make online queries while we only have a fixed dataset. In this section, we propose an empirical version of Algorithm 2. Especially, we use (1) randomized weighted trajectories to enlarge the query space, (2) an optimistic bonus to account for the uncertainty over reward functions, and (3) ensembles of $Q$-functions to select most explorative trajectories. The detailed algorithm is shown in Algorithm 1.

**Randomized Weighted Trajectories.** In offline PbRL, we can't query for new trajectories as in online RL. Limited query space can make the problem exponentially hard Zhan et al. (2023). As shown in Figure 2, A rescue is to add temporal weights to existing trajectories for fine-grained

comparison for better query efficiency. Since we don't know the critical part of the trajectory, empirically, we augment each trajectory with $L$ set of random Gaussian weights clipped to $[-1, 1]$ to control the variance. Then, the augmented dataset becomes

$$\mathcal{D}_{\text{off}}^{\text{aug}} = \{(w_i^{(\ell)}, \tau_i)_{\ell=1}^{L}\}_{i=1}^{N}, \quad w_h^{(\ell)} \sim \text{clip}(\mathcal{N}(0, 1), -1, 1) \tag{11}$$

**Optimistic Bonus.** Since it is computationally challenging to compute the reward confidence set and the near-optimal confidence set, we instead add an optimistic bonus to account for the uncertainty over reward functions. First, we train an ensemble of $M$ reward functions parameterized by $\theta_i$ via preference transformer (Kim et al., 2023) and cross-entropy loss as in Equation 6. Then we calculate its optimistic variant $\widetilde{r}_i$ as

$$\widetilde{r}_i(s, a) = r_{\theta_i}(s, a) + k\sigma(s, a), \tag{12}$$

where $\sigma(s, a) = \sqrt{\frac{1}{M} \sum_{i=1}^{M} (r_{\theta_i}(s, a) - \mu(s, a))^2}$ and $\mu(s, a) = \frac{1}{M} \sum_{i=1}^{M} r_{\theta_i}(s, a)$.

**Maximizing Value Difference.** To encourage exploration, we choose augmented trajectories $(\tau^{k,1}, \tau^{k,2})$ at each round via the following criteria, as indicated by Equation 9

$$\underset{(w_1, \tau_1), (w_2, \tau_2) \in \mathcal{D}_{\text{off}}^{\text{aug}}}{\text{argmax}} \underset{i,j \in [N]}{\text{argmax}} \left| \left( \widetilde{V}_i(w_1, \tau_1) - \widetilde{V}_j(w_2, \tau_2) \right) - \left( \widetilde{V}_i(w_2, \tau_2) - \widetilde{V}_j(w_1, \tau_1) \right) \right|, \tag{13}$$

where $\widetilde{V}_i(w_1, \tau_1) = \sum_{h=1}^{H} w_{1,h} \cdot \widetilde{r}_i(s_{h,1}, a_{h,1})$.

**Policy Extraction.** After querying all the preferences, we train an optimistic value function $Q_\psi$ and $V_\psi$ with IQL (Kostrikov et al., 2021) by minimizing the following loss:

$$L_Q(\psi) = \mathbb{E}_{(s,a,s') \sim \mathcal{D}_{\text{off}}} \left[ L_2^\tau (\mu(s, a) + k\sigma(s, a) + \gamma V_\phi(s') - Q_\phi(s, a) \right],$$

$$L_V(\psi) = \mathbb{E}_{(s,a,s',a') \sim \mathcal{D}_{\text{off}}} \left[ L_2^\tau (Q_{\widehat{\phi}}(s', a') - V_\phi(s, a) \right], \tag{14}$$

where $L_2^\tau(\cdot)$ is the $\tau$-expectile loss. Using an expectile loss gives a conservative estimation for the value function, which aligns with using PEVI in Algorithm 1. Finally, we extract the policy $\pi_\xi$ via minimizing the following objective

$$L_\pi(\xi) = \mathbb{E}_{(s,a) \sim \mathcal{D}} [\exp(\alpha(Q_\phi(s, a) - V_\phi(s))) \log(\pi_\xi(a|s))]. \tag{15}$$

---

**Algorithm 1** OPRIDE: Offline Preference-Based Reinforcement Learning with In-Dataset Exploration, General MDP

1: **Input**: Unlabeled offline dataset $\mathcal{D}_{\text{off}} = \{\tau_n = \{(s_h^n, a_h^n)\}_{h=1}^{H}\}_{n=1}^{N}$, query budget $K$, ensemble number $N$
2: Initialized preference dataset $\mathcal{D}_{\text{pref}} \leftarrow \emptyset$.
3: Initialize augmented offline dataset $\mathcal{D}_{\text{off}}^{\text{aug}}$ with random weights in Equation 11.
4: **for** episode $k = 1, \cdots, K$ **do**
5:     Train $N$ ensembles of preference network $r_{\theta_i}$ with $\mathcal{D}_{\text{pref}}$ using cross entropy loss in Equation 6 and calculate its optimistic variant as in Equation 12.
6:     Select augmented trajectories $(w^{k,1}, \tau^{k,1}), (w^{k,2}, \tau^{k,2})$ that maximize the value difference according to Equation 13.
7:     Receive the preference $o_k$ between $(w^{k,1}, \tau^{k,1})$ and $(w^{k,2}, \tau^{k,2})$ and add it to the preference dataset, i.e.,

$$\mathcal{D}_{\text{pref}} \leftarrow \mathcal{D}_{\text{pref}} \cup \{((w^{k,1}, \tau^{k,1}), (w^{k,2}, \tau^{k,2}), o_k)\}.$$

8: **end for**
9: Train optimistic $Q$-function $Q_\psi$ with bonus as in Equation 14.
10: Extract policy $\pi_\xi$ via Equation 15.
11: **Output**: The learned policy $\pi_\xi$

| Task | OPRL | PT | PT+PDS | OPRIDE | True Reward |
|---|---|---|---|---|---|
| assembly-v2 | 10.1±0.5 | 10.2±0.7 | 12.8±0.6 | **23.7±7.7** | 18.3±6.9 |
| basketball-v2 | 11.7±10.2 | 80.7±0.1 | 78.7±2.0 | **84.2±0.2** | 87.3±0.5 |
| button-press-wall-v2 | 51.7±1.6 | 58.8±0.9 | 59.4±0.9 | **76.4±0.3** | 63.0±2.1 |
| box-close-v2 | 15.0±0.7 | 17.7±0.1 | 17.2±0.3 | **59.1±4.1** | 99.2±1.0 |
| coffee-push-v2 | 1.7±1.7 | 1.3±0.5 | 1.3±0.5 | **59.8±0.6** | 20.4±2.7 |
| disassemble-v2 | 8.4±0.8 | 6.0±0.4 | 7.6±0.2 | **31.7±2.8** | 44.7±9.0 |
| door-close-v2 | 61.2±1.3 | **65.1±10.1** | 62.4±8.7 | 60.8±5.1 | 79.1±2.3 |
| handle-pull-side-v2 | 11.8±5.3 | 0.1±0.0 | 0.1±0.1 | **40.1±0.2** | 29.5±6.5 |
| peg-insert-side-v2 | 3.5±1.8 | 16.8±0.1 | 12.4±1.4 | **61.9±0.6** | 73.5±1.1 |
| push-v2 | 10.6±1.5 | 16.7±5.0 | 1.8±0.4 | **76.0±0.2** | 1.8±0.4 |

Table 1: Performance of offline RL algorithm on the reward-labeled dataset with different preference reward learning methods on the Meta-World tasks with 10 queries. "True Reward" denotes the performance of offline RL algorithms under the original reward function of the dataset. Complete experimental results are shown in Appendix F.

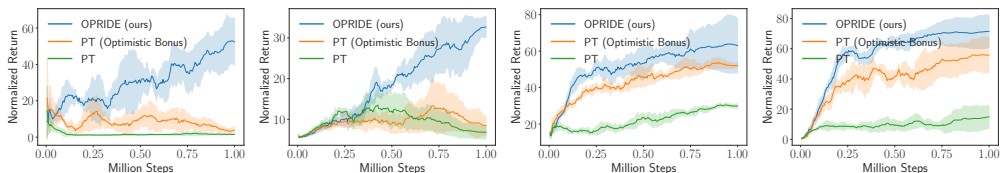

Figure 3: Modular ablation study on the Meta-World tasks. Tasks from left to right are `coffee-push-v2`,`disassemble-v2`,`hammer-v2`, and `push-v2`. $x$-axis denotes the offline training step and $y$-axis denotes the normalized offline performance.

## 4 EXPERIMENTS

In this section, we aim to answer the following questions: (1) How does our method perform on navigation and manipulation tasks? (2) Can our method improve query efficiency compared to other offline preference-based RL methods? (3) How effective is each part of the proposed method?

| Domain | Task | OPRL | PT | PT+PDS | OPRIDE | True Reward |
|---|---|---|---|---|---|---|
| Antmaze | umaze | 76.3±3.7 | 77.5±4.5 | 84.5±8.5 | **87.5±5.6** | 87.5±4.3 |
| | umaze-diverse | 72.5±3.4 | 68.0±3.0 | **78.0±6.0** | 73.1±2.4 | 62.2±4.1 |
| | medium-play | 0.0±0.0 | 63.5±0.5 | **72.5±6.5** | 62.2±2.0 | 73.8±4.4 |
| | medium-diverse | 0.0±0.0 | 63.5±4.5 | 58.0±4.0 | **69.4±5.2** | 68.1±10.1 |
| | large-play | 7.3±0.9 | 6.5±2.5 | 9.0±8.0 | **27.5±12.5** | 48.7±4.3 |
| | large-diverse | 6.9±2.4 | **23.5±0.5** | 8.5±2.5 | 21.5±1.5 | 44.3±4.4 |
| Average | | 27.1±1.7 | 50.4±2.5 | 51.7±5.9 | **56.8±4.8** | 64.1±5.2 |

Table 2: Experiments between several baselines and OPRIDE on the Antmaze tasks with 10 queries.

### 4.1 SETTING

To answer the above questions, we perform empirical evaluations on the Meta-World (Yu et al., 2019) and Antmaze on the D4RL benchmark (Fu et al., 2020). For the meta-world tasks, we create

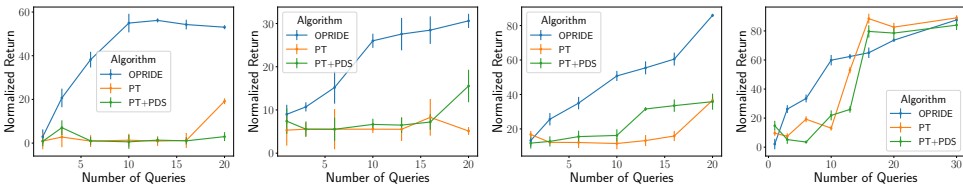

Figure 4: Performance of offline preference-based RL algorithms with different queries. Tasks from left to right are `coffee-push-v2,disassemble-v2,hammer-v2,` and `push-v2.`

a challenging benchmark, which is created by adding random action noises and $\epsilon$-greedy random actions to the scripted policies.

**Baselines:** We adopt Offline Preference-based Reward Learning (OPRL; Shin et al., 2023), Preference Transformer (PT; Kim et al., 2023) and Provable Data Sharing (PT+PDS; Hu et al., 2023) as the baselines algorithms. Specifically, OPRL and PT, respectively, use disagreement mechanisms and transformer architecture to learn the reward model. As for PT+PDS, we use multiple seeds to train PT, and then use PDS technology to generate the final reward value. Please refer to Appendix H for the experimental details.

### 4.2 EXPERIMENTAL RESULTS

**Answer to Question 1:** We conducted a comprehensive comparative analysis of OPRIDE against several baseline methods, utilizing both Meta-World and Antmaze tasks as our testing grounds. Specifically, all offline preference-based reinforcement learning (RL) methods were trained with 10 queries to establish the reward model. Subsequently, all algorithms employed the IQL algorithm for subsequent offline training. It's worth noting that our method also adopts the same transformer architecture as PT. The experimental results, presented in Table 1 and Table 2, clearly demonstrate the superior performance of our approach.

**Answer to Question 2:** We performed ablation studies to assess the quality of the reward model. In particular, we varied the number of queries used to train the reward model. The results presented in Figure 4 demonstrate that OPRIDE significantly outperforms the baselines across various query quantities. It's worth noting that the baselines require multiple times the number of queries to achieve performance on par with OPRIDE.

**Answer to Question 3:** We conducted module ablation studies using Meta-World tasks. Given that our method is based on PT (Preference-based Training), we introduced an optimistic bonus to PT in order to assess the efficacy of our query selection and dual policy. The results depicted in Figure 3 demonstrate the value of the dual mechanism in specific tasks.

## 5 CONCLUSION

This paper proposes a new methodology, i.e., in-dataset exploration, to allow query-efficient offline preference-based RL. Our proposed algorithm, OPRIDE, conducts principled in-dataset exploration by weighted trajectory queries, and a principled exploration strategy deals with pairwise queries and a balance between optimism and pessimism. Our method has provable guarantees, while our practical variant achieves strong empirical performance on various tasks. Compared to prior methods, our method significantly reduces the amount of queries required. Overall, our method provides a promising and principled way to reduce queries required from human labelers.

## 6 REPRODUCIBILITY

A comprehensive description of our algorithm implementation is provided in Section 3.2. The hyper-parameter configurations are detailed in Appendix H. The code necessary to reproduce OPRIDE are provided in our supplementary materials. Our theoretical findings are expounded upon in Section 3.1, with a detailed proof presented in Appendix C.

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

# A  DETAILS OF OPRIDE IN LINEAR MDPS

Here we provide an detailed description of our OPRIDE algorithm in linear MDPs with online queries in Algorithm 2.

---

**Algorithm 2** OPRIDE : Offline PbRL with In-Dataset Exploration, Linear MDP

---

1: **Input**: Unlabeled offline dataset $\mathcal{D}_{\text{off}} = \{\tau_n = \{(s_h^n, a_h^n)\}_{h=1}^H\}_{n=1}^N$, query budget $K$
2: Initialized preference dataset $\mathcal{D}_{\text{pref}} \leftarrow \emptyset$.
3: **for** $k = 1, \cdots, K$ **do**
4:    Calculate confidence set $\mathcal{C}_k(\delta)$ for reward function based on $\mathcal{D}_{\text{pref}}$ with Equation 7.
5:    Calculate pessimistic value function $\widehat{V}_\theta(\cdot)$ using Algorithm 3 for each $\theta \in \mathcal{C}_k(\delta)$.
6:    Construct the near-optimal policy set $\Pi_k$ using Equation 8.
7:    Select explorative policies $\pi^{k,1}, \pi^{k,2}$ within $\Pi_k$ based on Equation 9.
8:    Sample trajectories $\tau^{k,1}, \tau^{k,2}$ with selected policy $\pi^{k,1}, \pi^{k,2}$.
9:    Receive the preference $o_k$ between $\tau^{k,1}$ and $\tau^{k,2}$ and add it to the preference dataset

$$\mathcal{D}_{\text{pref}} \leftarrow \mathcal{D}_{\text{pref}} \cup \{(\tau^{k,1}, \tau^{k,2}, o_k)\}.$$

10: **end for**
11: **Output**: The average policy $\bar{\pi} = 1/(2K) \cdot \sum_{k=1}^K (\pi^{k,1} + \pi^{k,2})$.

---

## A.1  PROJECTED MLE ESTIMATOR

In the construction of the confidence set, we use the projected MLE estimator $\widehat{\theta}_k$ as the center of the confidence set. To obtain the projected MLE estimator, we first construct the standard MLE estimator by minimizing the following negative log-liklihood or cross entropy

$$\mathcal{L}_\lambda(\theta) = -\sum_{n=1}^N \left( o_n \log(\sigma(\langle \phi(\tau_n^1, \tau_n^2), \theta \rangle)) + (1 - o_n) \log(1 - \sigma(\langle \phi(\tau_n^1, \tau_n^2), \theta \rangle)) \right) - \frac{\lambda}{2} \|\theta\|_2^2, \quad (16)$$

where $\lambda$ is the coefficient for the regularization. Then we calculate the projected MLE estimator by optimizing the following objective

$$\widehat{\theta}_k = \underset{\|\theta\| \leq W}{\text{argmin}} \|g_k(\theta) - g_k(\widehat{\theta}_k^{\text{MLE}})\|_{\Sigma_k^{-1}} \quad (17)$$

$$(18)$$

where

$$\Sigma_k = \lambda I + \sum_{i=1}^{k-1} \left( \phi(\tau_i^1) - \phi(\tau_i^2) \right) \left( \phi(\tau_i^1) - \phi(\tau_i^2) \right)^\top,$$

$$g_k(\theta) = \sum_{i=1}^{k-1} \sigma(\langle \phi(w_i^1, \tau_i^1) - \phi(w_i^2, \tau_i^2), \theta \rangle) \left( \phi(w_i^1, \tau_i^1) - \phi(w_i^2, \tau_i^2) \right) + \lambda \theta,$$

and $\phi(\tau) = \sum_{h=1}^H \phi(s_h^\tau, a_h^\tau)$.

# B DETAILS OF PESSIMISTIC VALUE ITERATION (PEVI; JIN ET AL., 2021)

In this section, we consider *pessimistic value iteration* (PEVI; Jin et al., 2021) as the backbone algorithm, described in Algorithm 3. It is a representative model-free offline algorithm with theoretical guarantees. PEVI uses negative bonus $\Gamma(\cdot, \cdot)$ over standard $Q$-value estimation $\widehat{Q}(\cdot, \cdot) = (\widehat{\mathbb{B}}\widehat{V})(\cdot)$ to reduce potential bias due to finite data, where $\widehat{\mathbb{B}}$ is some empirical estimation of $\mathbb{B}$ from dataset $\mathcal{D}$. We use the following notion of $\xi$-uncertainty quantifier as follows to formalize the idea of pessimism.

**Definition 4** ($\xi$-Uncertainty Quantifier). *We say $\Gamma : \mathcal{S} \times \mathcal{A} \to \mathbb{R}$ is a $\xi$-uncertainty quantifier for $\widehat{\mathbb{B}}$ and $\widehat{V}$ if with probability $1 - \xi$, for all $(s, a) \in \mathcal{S} \times \mathcal{A}$,*

$$\left| (\widehat{\mathbb{B}}\widehat{V})(s, a) - (\mathbb{B}\widehat{V})(s, a) \right| \leq \Gamma(s, a). \tag{19}$$

In linear MDPs, we can construct $\widehat{\mathbb{B}}\widehat{V}$ and $\Gamma$ based on $\mathcal{D}$ as follows, where $\widehat{\mathbb{B}}\widehat{V}$ is the empirical estimation for $\mathbb{B}\widehat{V}$. For a given dataset $\mathcal{D} = \{(s_\tau, a_\tau, r_\tau)\}_{\tau=1}^N$, we define the empirical mean squared Bellman error (MSBE) as

$$M(w) = \sum_{\tau=1}^N \left( r_\tau + \gamma \widehat{V}(s_{\tau+1}) - \phi(s_\tau, a_\tau)^\top w \right)^2 + \lambda \|w\|_2^2$$

Here $\lambda > 0$ is the regularization parameter. Note that $\widehat{w}$ has the closed form

$$\widehat{w} = \Lambda^{-1} \left( \sum_{\tau=1}^N \phi(s_\tau, a_\tau) \cdot \left( r_\tau + \gamma \widehat{V}(s_{\tau+1}) \right) \right),$$

$$\text{where } \Lambda = \lambda I + \sum_{\tau=1}^N \phi(s_\tau, a_\tau)\phi(s_\tau, a_\tau)^\top. \tag{20}$$

Then we simply let

$$\widehat{\mathbb{B}}\widehat{V} = \langle \phi, \widehat{w} \rangle. \tag{21}$$

Meanwhile, we construct $\Gamma$ based on $\mathcal{D}$ as

$$\Gamma(s, a) = \beta \cdot \left( \phi(s, a)^\top \Lambda^{-1} \phi(s, a) \right)^{1/2}. \tag{22}$$

Here $\beta > 0$ is the scaling parameter. The overall PEVI algorithm is summarized in Algorithm 3. Also, we provide a variant of PEVI algorithm that evaluates a given policy, as shown in Algorithm 4.

We first characterize the quality of the dataset with the notion of coverage coefficient (Uehara & Sun, 2021), defined as below.

**Definition 5.** *The coverage coefficient $C^\dagger$ of a dataset $\mathcal{D} = \{(s_\tau, a_\tau, r_\tau)\}_{\tau=1}^N$ is defined as*

$$C^\dagger = \sup_C \left\{ \frac{1}{N} \cdot \sum_{\tau=1}^N \phi(s_\tau, a_\tau)\phi(s_\tau, a_\tau)^\top \succeq C \cdot \mathbb{E}_{\pi^*} \left[ \phi(s_t, a_t)\phi(s_t, a_t)^\top \mid s_0 = s \right], \forall s \in \mathcal{S} \right\}, \tag{23}$$

The coverage coefficient $C^\dagger$ is common in offline RL literature (Uehara & Sun, 2021; Jin et al., 2021; Rashidinejad et al., 2021), which represents the maximum ratio between the density of empirical state-action distribution and the density induced from the optimal policy. Intuitively, it represents

the quality of the dataset. For example, the `expert` dataset has a high coverage ratio while the `random` dataset may have a low ratio.

---

**Algorithm 3** Pessmistic Value Iteration Jin et al. (2021)

---

1: **Input**: Offline Dataset $\mathcal{D}_{\text{off}} = \{\tau_k = \{(s_h^k, a_h^k)\}_{h=1}^H\}_{k=1}^K$, reward function $r_\theta$.
2: Set $\widehat{V}_{H+1}(\cdot) = 0$
3: **for** episode $h = H, \cdots, 1$ **do**
4:     Set $\Lambda_h \leftarrow \sum_{\tau=1}^K \phi(s_h^\tau, a_h^\tau)\phi^\top(s_h^\tau, a_h^\tau) + \lambda \cdot I$.
5:     Set $\widehat{w}_h \leftarrow \Lambda_h^{-1}(\sum_{\tau=1}^K \phi(s_h^\tau, a_h^\tau) \cdot (r_\theta(s_h^\tau, a_h^\tau) + \widehat{V}_{h+1}(s_{h+1}^\tau)))$.
6:     Set $\Gamma_h(\cdot, \cdot) \leftarrow \beta \cdot (\phi(\cdot, \cdot)^\top \Lambda_h^{-1} \phi(\cdot, \cdot))^{1/2}$.
7:     Set $\widehat{Q}_h(\cdot, \cdot) \leftarrow \min\{\phi^\top(\cdot, \cdot)\widehat{w}_h - \Gamma(\cdot, \cdot), H - h + 1\}^+$.
8:     Set $\widehat{\pi}(\cdot|\cdot) \leftarrow \operatorname{argmax}_{\pi_h}\langle\widehat{Q}_h(\cdot, \cdot), \pi_h(\cdot|\cdot)\rangle_{\mathcal{A}}$.
9:     Set $\widehat{V}_h \leftarrow \langle\widehat{Q}_h(\cdot, \cdot), \widehat{\pi}_h(\cdot|\cdot)\rangle_{\mathcal{A}}$.
10: **end for**
11: **Output**: $\{\widehat{\pi}_h\}_{h=1}^H$.

---

---

**Algorithm 4** Pessmistic Policy Evaluation

---

1: **Input**: Offline Dataset $\mathcal{D}_{\text{off}} = \{\tau_k = \{(s_h^k, a_h^k)\}_{h=1}^H\}_{k=1}^K$, reward function $r_\theta$, policy $\pi(\cdot|\cdot)$.
2: Set $\widehat{V}_{H+1}(\cdot) = 0$
3: **for** episode $h = H, \cdots, 1$ **do**
4:     Set $\Lambda_h \leftarrow \sum_{\tau=1}^K \phi(s_h^\tau, a_h^\tau)\phi^\top(s_h^\tau, a_h^\tau) + \lambda \cdot I$.
5:     Set $\widehat{w}_h \leftarrow \Lambda_h^{-1}(\sum_{\tau=1}^K \phi(s_h^\tau, a_h^\tau) \cdot (r_\theta(s_h^\tau, a_h^\tau) + \widehat{V}_{h+1}(s_{h+1}^\tau)))$.
6:     Set $\Gamma_h(\cdot, \cdot) \leftarrow \beta \cdot (\phi(\cdot, \cdot)^\top \Lambda_h^{-1} \phi(\cdot, \cdot))^{1/2}$.
7:     Set $\widehat{Q}_h(\cdot, \cdot) \leftarrow \min\{\phi^\top(\cdot, \cdot)\widehat{w}_h - \Gamma(\cdot, \cdot), H - h + 1\}^+$.
8:     Set $\widehat{V}_h \leftarrow \langle\widehat{Q}_h(\cdot, \cdot), \pi_h(\cdot|\cdot)\rangle_{\mathcal{A}}$.
9: **end for**
10: **Output**: $\{\widehat{V}_h\}_{h=1}^H$.

---

## C  PROOF OF THEOREM 3

*Proof.* For any policy $\widetilde{\pi} = \operatorname{greedy}\left(\widehat{V}_{\widetilde{\theta}}\right), \widetilde{\theta} \in \mathcal{C}_k(\delta)$, we have

$$
\begin{aligned}
V_{\theta^\star}^{\pi^\star} - V_{\theta^\star}^{\widetilde{\pi}} &= V_{\theta^\star}^{\pi^\star} - \widehat{V}_{\theta^\star}^{\pi^\star} + \widehat{V}_{\theta^\star}^{\pi^\star} - \widehat{V}_{\widetilde{\theta}}^{\pi^\star} + \widehat{V}_{\widetilde{\theta}}^{\pi^\star} - \widehat{V}_{\widetilde{\theta}}^{\widetilde{\pi}} + \widehat{V}_{\widetilde{\theta}}^{\widetilde{\pi}} - \widehat{V}_{\theta^\star}^{\widetilde{\pi}} + \widehat{V}_{\theta^\star}^{\widetilde{\pi}} - V_{\theta^\star}^{\widetilde{\pi}} \\
&\leq V_{\theta^\star}^{\pi^\star} - \widehat{V}_{\theta^\star}^{\pi^\star} + \widehat{V}_{\theta^\star}^{\pi^\star} - \widehat{V}_{\widetilde{\theta}}^{\pi^\star} + \widehat{V}_{\widetilde{\theta}}^{\pi^\star} - \widehat{V}_{\widetilde{\theta}}^{\widetilde{\pi}} + \widehat{V}_{\widetilde{\theta}}^{\widetilde{\pi}} - \widehat{V}_{\theta^\star}^{\widetilde{\pi}} + 0 \\
&\leq V_{\theta^\star}^{\pi^\star} - \widehat{V}_{\theta^\star}^{\pi^\star} + \widehat{V}_{\theta^\star}^{\pi^\star} - \widehat{V}_{\widetilde{\theta}}^{\pi^\star} + 0 \qquad\qquad + \widehat{V}_{\widetilde{\theta}}^{\widetilde{\pi}} - \widehat{V}_{\theta^\star}^{\widetilde{\pi}} \\
&\leq V_{\theta^\star}^{\pi^\star} - \widehat{V}_{\theta^\star}^{\pi^\star} + \operatorname*{argmax}_{\theta_1, \theta_2 \in \mathcal{C}_k(\delta)} \left(\widehat{V}_{\theta_1}^{\pi^\star} - \widehat{V}_{\theta_2}^{\pi^\star} + \widehat{V}_{\theta_2}^{\widetilde{\pi}} - \widehat{V}_{\theta_1}^{\widetilde{\pi}}\right) \\
&\leq V_{\theta^\star}^{\pi^\star} - \widehat{V}_{\theta^\star}^{\pi^\star} + \operatorname*{argmax}_{\theta_1, \theta_2 \in \mathcal{C}_k(\delta)} \left(\widehat{V}_{\theta_1}^{\widetilde{\pi}^{k,1}} - \widehat{V}_{\theta_2}^{\widetilde{\pi}^{k,1}} + \widehat{V}_{\theta_2}^{\widetilde{\pi}^{k,2}} - \widehat{V}_{\theta_1}^{\widetilde{\pi}^{k,2}}\right),
\end{aligned}
\tag{24}
$$

which hold with probability $1 - \delta$. The first inequality follows from the pessimistic property of $\widehat{V}$, the second inequality follows from the fact that $\widetilde{\pi} = \operatorname{greedy}(\widehat{V}_{\widetilde{\theta}})$. The third inequality holds since $\widetilde{\theta}, \theta^\star \in \mathcal{C}_k(\delta)$ and the last inequality follows from the definition of $\widetilde{\pi}^{k,1}, \widetilde{\pi}^{k,2}$.

Following Lemma 6, we have for all policy $\pi$ and reward function $\theta$,

$$|V_\theta^\pi - \widehat{V}_\theta^\pi| \leq c \cdot \sqrt{\frac{C^\dagger d^3 H^5 \zeta_1}{N}} := \epsilon.$$

Then we have

$$V_{\theta^\star}^{\pi^\star} - V_{\theta^\star}^{\widetilde{\pi}} \leq V_{\theta^\star}^{\pi^\star} - \widehat{V}_{\theta^\star}^{\pi^\star} + \underset{\theta_1, \theta_2 \in \mathcal{C}_k(\delta)}{\operatorname{argmax}} \left( \widehat{V}_{\theta_1}^{\widetilde{\pi}^{k,1}} - \widehat{V}_{\theta_2}^{\widetilde{\pi}^{k,1}} + \widehat{V}_{\theta_2}^{\widetilde{\pi}^{k,2}} - \widehat{V}_{\theta_1}^{\widetilde{\pi}^{k,2}} \right) \tag{25}$$

$$\leq \epsilon + \underset{\theta_1, \theta_2 \in \mathcal{C}_k(\delta)}{\operatorname{argmax}} \left( \left( V_{\theta_1}^{\widetilde{\pi}^{k,1}} - V_{\theta_2}^{\widetilde{\pi}^{k,1}} + V_{\theta_2}^{\widetilde{\pi}^{k,2}} - V_{\theta_1}^{\widetilde{\pi}^{k,2}} \right) + \left( \widehat{V}_{\theta_1}^{\widetilde{\pi}^{k,1}} - V_{\theta_1}^{\widetilde{\pi}^{k,1}} \right) \right. \tag{26}$$

$$\left. + \left( \widehat{V}_{\theta_2}^{\widetilde{\pi}^{k,1}} - V_{\theta_2}^{\widetilde{\pi}^{k,1}} \right) + \left( \widehat{V}_{\theta_2}^{\widetilde{\pi}^{k,2}} - V_{\theta_2}^{\widetilde{\pi}^{k,2}} \right) + \left( \widehat{V}_{\theta_1}^{\widetilde{\pi}^{k,2}} - V_{\theta_1}^{\widetilde{\pi}^{k,2}} \right) \right) \tag{27}$$

$$\leq \epsilon + \underset{\theta_1, \theta_2 \in \mathcal{C}_k(\delta)}{\operatorname{argmax}} \left( \left( V_{\theta_1}^{\widetilde{\pi}^{k,1}} - V_{\theta_2}^{\widetilde{\pi}^{k,1}} + V_{\theta_2}^{\widetilde{\pi}^{k,2}} - V_{\theta_1}^{\widetilde{\pi}^{k,2}} \right) + 4\epsilon \right) \tag{28}$$

$$= 5\epsilon + \underset{\theta_1, \theta_2 \in \mathcal{C}_k(\delta)}{\operatorname{argmax}} \left( V_{\theta_1}^{\widetilde{\pi}^{k,1}} - V_{\theta_2}^{\widetilde{\pi}^{k,1}} + V_{\theta_2}^{\widetilde{\pi}^{k,2}} - V_{\theta_1}^{\widetilde{\pi}^{k,2}} \right). \tag{29}$$

Consider the online preference-based regret as

$$\operatorname{Reg}(T) = \frac{1}{2} \sum_{k=1}^{K} \left( V^{\pi^\star} - V^{\widetilde{\pi}^{k,1}} + V^{\pi^\star} - V^{\widetilde{\pi}^{k,2}} \right), \tag{30}$$

we have

$$\text{Reg}(T)$$

$$\leq \sum_{k=1}^{K} \underset{\theta_1, \theta_2 \in \mathcal{C}_k(\delta)}{\text{argmax}} \left( V_{\theta_1}^{\widetilde{\pi}^{k,1}} - V_{\theta_2}^{\widetilde{\pi}^{k,1}} + V_{\theta_2}^{\widetilde{\pi}^{k,2}} - V_{\theta_1}^{\widetilde{\pi}^{k,2}} \right) + 5K\epsilon$$

$$= \sum_{k=1}^{K} \underset{\theta_1, \theta_2 \in \mathcal{C}_k(\delta)}{\text{argmax}} \left\{ \left( V_{\theta_1}(\tau^{k,1}) - V_{\theta_2}(\tau^{k,1}) + V_{\theta_2}(\tau^{k,2}) - V_{\theta_1}(\tau^{k,2}) \right) + \right.$$

$$+ (V_{\theta_1}^{\widetilde{\pi}^{k,1}} - V_{\theta_1}(\tau^{k,1})) - (V_{\theta_2}^{\widetilde{\pi}^{k,1}} - V_{\theta_2}(\tau^{k,1}))$$

$$\left. + (V_{\theta_1}^{\widetilde{\pi}^{k,2}} - V_{\theta_1}(\tau^{k,2})) - (V_{\theta_2}^{\widetilde{\pi}^{k,2}} - V_{\theta_2}(\tau^{k,2})) \right\} + 5K\epsilon$$

$$\leq \sum_{k=1}^{K} \underset{\theta_1, \theta_2 \in \mathcal{C}_k(\delta)}{\text{argmax}} \left( V_{\theta_1}(\tau^{k,1}) - V_{\theta_2}(\tau^{k,1}) + V_{\theta_2}(\tau^{k,2}) - V_{\theta_1}(\tau^{k,2}) \right)$$

$$+ 5K\epsilon + 16\sqrt{H^2 K \log\left(\frac{4}{\delta}\right)}$$

$$= \sum_{k=1}^{K} \underset{\theta_1, \theta_2 \in \mathcal{C}_k(\delta)}{\text{argmax}} \ (\phi(\tau^{k,1}) - \phi(\tau^{k,2}))^\top (\theta_1 - \theta_2) + 5K\epsilon + 16\sqrt{H^2 K \log\left(\frac{4}{\delta}\right)}$$

$$\leq \sum_{k=1}^{K} \underset{\theta_1, \theta_2 \in \mathcal{C}_k(\delta)}{\text{argmax}} \ \|\phi(\tau^{k,1}) - \phi(\tau^{k,2})\|_{\Sigma_k^{-1}} \|\theta_1 - \theta_2\|_{\Sigma_k} + 5K\epsilon + 16\sqrt{H^2 K \log\left(\frac{4}{\delta}\right)}$$

$$\leq B(\delta) \sum_{k=1}^{K} \|\phi(\tau^{k,1}) - \phi(\tau^{k,2})\|_{\Sigma_k^{-1}} + 5K\epsilon + 16\sqrt{H^2 K \log\left(\frac{4}{\delta}\right)}$$

$$\leq B(\delta) \sqrt{K \sum_{k=1}^{K} \|\phi(\tau^{k,1}) - \phi(\tau^{k,2})\|_{\Sigma_k^{-1}}^2} + 5K\epsilon + 16\sqrt{H^2 K \log\left(\frac{4}{\delta}\right)}$$

$$\leq B(\delta) \sqrt{2Kd \log(1 + \frac{KH}{d})} + 5K\epsilon + 16\sqrt{H^2 K \log\left(\frac{4}{\delta}\right)}$$

$$\leq c \cdot \sqrt{d^2 H^2 K \zeta_2'} + 5K\epsilon + 16\sqrt{H^2 K \log\left(\frac{4}{\delta}\right)}$$

$$\leq c' \cdot \sqrt{d^2 H^2 K \zeta_2} + 5K\epsilon,$$

where $\zeta_2' = \log\left(T(1+2T)/\delta\right) \log(1 + \frac{KH}{d})$ and $\zeta_2 = \zeta_2' + \log\left(4/\delta\right)$. The first inequality follows from Equation 29. The second inequality follows from Azuma-Hoeffding's inequality and the fact that $V_\theta(\tau) - V_\theta^\pi$ is a martingale when $\tau \sim \pi$. Please refer to Cai et al. (2020) for a detailed derivation. The third inequality follows from Cauchy-Schwarz inequality and the fourth and fifth inequalties follows from Lemma 8 and Lemma 9. The last inequality combines the first term and the third term together.

Finally, follows a standard argument for regret to PAC conversion (Jin et al., 2018), we can show that with a high probability, the suboptimality of average policy $\bar{\pi}$ generated by Algorithm 2 is upper bounded by

$$\text{SubOpt}(\bar{\pi}) \leq c' \cdot \left( \epsilon + H \cdot \sqrt{\frac{d^2 H^2 \zeta_2}{K}} \right) = c_0 \cdot \sqrt{\frac{d^3 H^5 C^\dagger \zeta_1}{N}} + c_1 \cdot \sqrt{\frac{d^2 H^4 \zeta_2}{K}}.$$

$\square$

## D    AUXILIARY LEMMAS

**Lemma 6** (Suboptimality of PEVI and PPE). *For the output $\widehat{\pi}$ of Algorithm 3, the following inequality*

$$\text{SubOpt}(\widehat{\pi}; s) \leq c \cdot \sqrt{\frac{C^{\dagger} d^3 H^5 \zeta_1}{N}}, \tag{31}$$

*holds for all $s \in \mathcal{S}$ with probability $1 - \delta$, where $c$ is an absolute constant and $\zeta_1 = \log(4dN/\delta)$. Similarly, for any policy $\pi$, the output $\widehat{V}(\pi)$ of Algorithm 4 satifies*

$$0 \leq V_1(\pi) - \widehat{V}_1(\pi) \leq c \cdot \sqrt{\frac{C^{\dagger} d^3 H^5 \zeta_1}{N}}, \tag{32}$$

*for all $s \in \mathcal{S}$ with probability $1 - \delta$,*

*Proof.* Following Corollary 4.5 in Jin et al. (2021), we have Equation 31 immediately. Equation 32 is a simple extension of Corollary 4.5 in Jin et al. (2021) and we omit the proof for simplicity. $\square$

**Lemma 7.** *[Lemma 1 from Faury et al. (2020)] Let $\delta \in (0, 1]$ and define the event that $\theta^{\star}$ is in the confidence interval $\mathcal{C}_k(\delta)$ with $\beta_k(\delta) = \sqrt{\lambda} + \sqrt{\log(1/\delta) + 2d \log\left(1 + \frac{k}{\lambda d}\right)}$ for all $k \in \mathbb{N}$:*

$$\mathcal{E}_{\delta} = \{\forall k \geq 1, \theta^{\star} \in \mathcal{C}_k(\delta)\}.$$

*Then $\mathbb{P}(\mathcal{E}_{\delta}) \geq 1 - \delta$.*

*Proof.* Please refer to Faury et al. (2020) for a detailed proof. $\square$

**Lemma 8.** *With probability $1 - \delta \log(T)$, the following inequality holds for all $\delta \leq 1/e$.*

$$\|\theta_1 - \theta_2\|_{\Sigma_k} \leq B(\delta), \tag{33}$$

*where $B(\delta) = (8 + 40H)\sqrt{d \log\left(T(1 + 2T)/\delta\right)} + \sqrt{H} \leq c \cdot \sqrt{dH^2 \log\left(T(1 + 2T)/\delta\right)}$.*

*Proof.* Please refer to Corollary 1 in Pacchiano et al. (2021) for a detailed proof. $\square$

**Lemma 9** (Lemma 19.4 of Lattimore & Szepesvári (2020)). *Let $\lambda = 1$. Consider the sequence $v_1, v_2, \ldots, v_T \in \mathbb{R}^d$ such that $\|v_t\| \leq 1$ for all $t \in [T]$. Define $V_t = \lambda I + \sum_{s=1}^{t-1} v_s v_s^{\top}$, then we have*

$$\sum_{t \in [T]} \|v_t\|_{V_t^{-1}}^2 \leq 2d \log\left(1 + \frac{T}{d}\right). \tag{34}$$

*Proof.* Since $\lambda = 1$, we have $\|v_t\|_{V_t^{-1}}^2 \leq 1$, therefore

$$\sum_{t \in [T]} \|v_t\|_{V_t^{-1}}^2 \leq 2 \sum_{t \in [T]} \log\left(1 + \|v_t\|_{V_t^{-1}}^2\right)$$

$$= 2 \sum_{t \in [T]} \log\left(\frac{\det V_t}{\det V_{t-1}}\right)$$

$$= 2 \log\left(\frac{\det V_T}{\det V_0}\right)$$

$$\leq 2 \log\left(\frac{d\lambda + T}{d\lambda}\right)$$

$$= 2 \log\left(1 + \frac{T}{d\lambda}\right).$$

$\square$

# E   DETAILS OF WEIGHTED QUERIES EXPERIMENT ON DIDACTIC MDPS

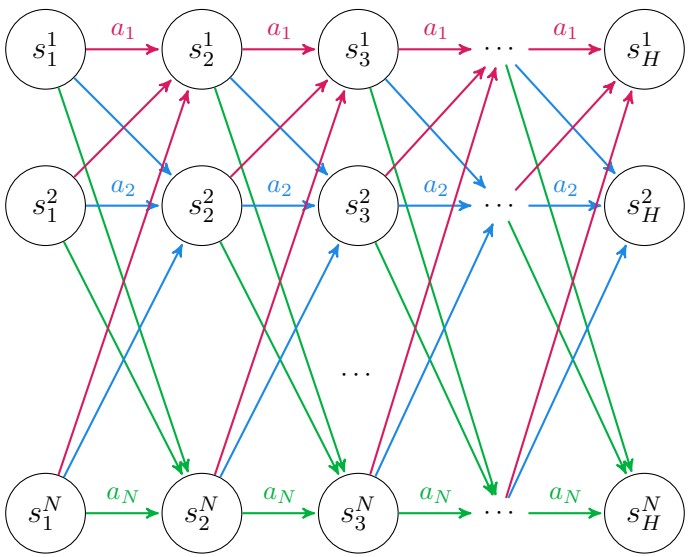

Figure 5: The didactic chain MDP. The reward is defined on the states and different colors denotes different actions. The transition is deterministic, where action $a_j$ leads to $s_{h+1}^j$ regardless of $s_h^i$ for all $i, j \in [N], h \in [H]$. The features for each state $\phi(s)$ are randomly generated from a Gaussian distribution.

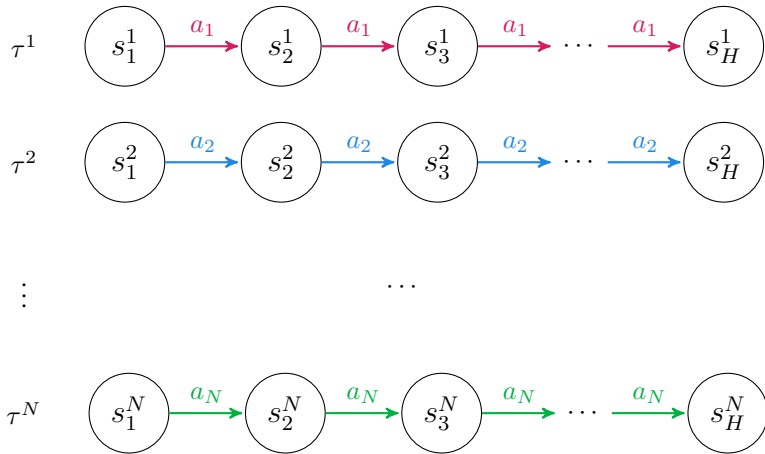

Figure 6: The didactic chain MDP with given dataset $\{\tau_i\}_{i=1}^N$. Each trajectory consist one and only one action. The trajectories covers the whole state space and do not intersect with each other.

To illustrate the importance of weighted queries, we consider the chain MDP as depicted in Figure 5 and Figure 6.

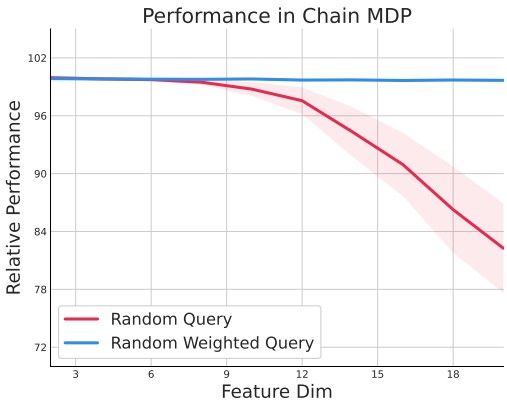 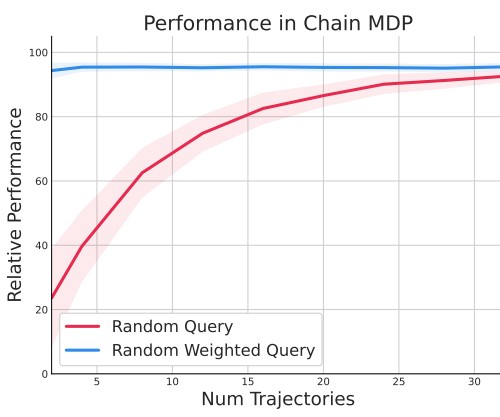

(a) The performance of weighted and unweighted random queries with various feature dimension.

(b) The performance of weighted and unweighted random queries with various numbers of trajectories in the dataset.

Figure 7: Results on the didactic chain MDP. Weighted queries has significant advantage when feature dimension is high or the amount of trajectories in the dataset is limited. The results are averaged over 100 runs.

**The MDP.** As shown in Figure 5, the MDP consist of $NH$ states and $N$ actions. The transition is deterministic, where $a_j$ leads to $s_{h+1}^j$ for any $s_h^i$ regardless of $i$ and $h$. The reward is defined on states only and $r(s) = \phi(s)^\top \theta$ for some feature map $\phi(s) \in \mathbb{R}^d$ and parameter $\theta \in \mathbb{R}^d$.

**Dataset.** As shown in Figure 6, the dataset consists of $N$ non-intersect trajectories $\{\tau_i\}_{i=1}^N$ where $\tau_i = \{(s_h^i, a_h^i)\}_{h=1}^H$. Each trajectory consist one and only one action. The trajectories covers the whole state space and do not intersect with each other.

**Feature Map.** The features $\phi(s) \in \mathbb{R}^d$ for each state are randomly generated from the standard Gaussian distribution, i.e., $\mathcal{N}(0,1)$.

**Methods and baselines.** We assume that the exact dynamics of the MDP is known. For standard queries, we uniformly sample pairs of trajectories from the dataset. For weighted queries, we add random weights $w_h^i \sim \text{clip}(\mathcal{N}(0,1), -1, 1)$ to each state $s_h^i$ in the sampled trajectory $\tau_i$. After querying the preference, we use standard logistic regression to learn the reward function, and estimate the optimal policy.

**Results.** We set $d = 16, N = 16, H = 32$ for the experiment. The performance of random standard queries and random weighted queries are shown in Figure 2 (d). We can see that weighted queries significantly outperforms unweighted counterpart in terms of efficiency. To investigate how parameters like number of trajectories in the dataset $N$, and feature dimension $d$ influence the performance of weighted queries, we also conduct the above experiment with different set of parameters. The result is shown in Figure 7. It is clear that weighted queries has significant advantage when feature dimension is high or the amount of trajectories in the dataset is limited.

# F ADDITION EXPERIMENTAL RESULTS

| Task | OPRL | PT | PT+PDS | OPRIDE | True Reward |
|---|---|---|---|---|---|
| assembly-v2 | 10.1±0.5 | 10.2±0.7 | 12.8±0.6 | **23.7±7.7** | 18.3±6.9 |
| basketball-v2 | 11.7±10.2 | 80.7±0.1 | 78.7±2.0 | **84.2±0.2** | 87.3±0.5 |
| bin-picking-v2 | **82.0±5.6** | 31.9±16.2 | 53.4±19.0 | 75.5±2.9 | 94.9±2.7 |
| button-press-wall-v2 | 51.7±1.6 | 58.8±0.9 | 59.4±0.9 | **76.4±0.3** | 63.0±2.1 |
| box-close-v2 | 15.0±0.7 | 17.7±0.1 | 17.2±0.3 | **59.1±4.1** | 99.2±1.0 |
| coffee-push-v2 | 1.7±1.7 | 1.3±0.5 | 1.3±0.5 | **59.8±0.6** | 20.4±2.7 |
| dial-turn-v2 | 43.0±8.4 | 67.8±3.7 | **74.3±1.8** | 58.6±7.7 | 72.9±2.6 |
| disassemble-v2 | 8.4±0.8 | 6.0±0.4 | 7.6±0.2 | **31.7±2.8** | 44.7±9.0 |
| door-close-v2 | 61.2±1.3 | **65.1±10.1** | 62.4±8.7 | 60.8±5.1 | 79.1±2.3 |
| door-unlock-v2 | **79.2±2.3** | 73.7±5.4 | 73.6±4.8 | 73.8±3.4 | 72.9±1.9 |
| drawer-open-v2 | 53.0±3.3 | **59.7±1.3** | 58.3±0.1 | 52.6±4.0 | 70.2±0.5 |
| faucet-close-v2 | **60.8±1.0** | 57.8±0.9 | 46.2±0.2 | 52.1±1.2 | 55.1±3.1 |
| hammer-v2 | 16.4±1.0 | 30.2±1.7 | 32.6±0.8 | **72.5±14.9** | 96.3±0.6 |
| hand-insert-v2 | 5.2±3.2 | 18.7±0.1 | **20.3±0.6** | 16.7±2.3 | 47.0±19.9 |
| handle-press-v2 | **28.7±4.0** | 27.9±0.2 | 28.2±0.2 | 26.3±0.1 | 96.7±0.4 |
| handle-pull-side-v2 | 11.8±5.3 | 0.1±0.0 | 0.1±0.1 | **40.1±0.2** | 29.5±6.5 |
| lever-pull-v2 | **63.2±10.4** | 49.2±3.7 | 51.7±0.1 | 36.3±6.9 | 39.7±0.1 |
| peg-insert-side-v2 | 3.5±1.8 | 16.8±0.1 | 12.4±1.4 | **61.9±0.6** | 73.5±1.1 |
| pick-place-v2 | 0.6±0.0 | 0.8±0.3 | 0.8±0.1 | 0.6±0.3 | 15.7±14.5 |
| plate-slide-v2 | **77.4±1.6** | 4.9±0.0 | 37.3±2.3 | 38.8±3.9 | 79.3±3.6 |
| push-v2 | 10.6±1.5 | 16.7±5.0 | 1.8±0.4 | **76.0±0.2** | 1.8±0.4 |
| push-back-v2 | 0.8±0.0 | 1.1±0.4 | 1.1±0.1 | **21.0±1.9** | 55.2±10.1 |
| push-wall-v2 | 7.4±4.2 | 74.8±14.4 | 3.4±0.9 | **81.8±5.8** | 90.8±0.1 |
| reach-v2 | 63.5±2.9 | 82.0±0.8 | 84.3±0.9 | **86.0±0.4** | 87.3±1.4 |
| shelf-place-v2 | 12.3±1.6 | 0.0±0.0 | 1.6±1.6 | **13.2±0.1** | 12.3±1.7 |
| soccer-v2 | 34.3±4.0 | 51.3±4.1 | 41.5±11.9 | **62.3±0.8** | 70.5±15.6 |
| stick-pull-v2 | **28.0±14.3** | 15.3±0.8 | 0.3±0.2 | 24.3±2.0 | 25.8±8.7 |
| sweep-into-v2 | 37.1±13.9 | 9.8±0.2 | 9.2±0.1 | **70.5±0.3** | 77.5±0.1 |
| sweep-v2 | 6.8±1.8 | 8.0±0.4 | 8.0±0.1 | **32.0±5.7** | 85.8±0.4 |
| window-close-v2 | 41.2±3.1 | **47.5±1.0** | 39.8±10.0 | 44.1±3.3 | 27.1±3.6 |
| Average | 32.5±3.98 | 36.0±2.2 | 33.6±3.4 | **46.3±4.5** | 55.2±5.6 |

Table 3: Experiments between several baselines and OPRIDE on the Meta-World tasks with 10 queries. We select 30 representative tasks from 50 Meta-World MT1 tasks for evaluating. The experimental results are averaged with five random seeds.

# G ADDITION ABLATION STUDIES

We conducted additional ablation studies for our algorithm. Specifically, we name PT with optimistic bonus term (Equation (12)) as PT (OB). We name PT with optimistic bonus and value difference term (Equation (13)) as PT (OB + VD). We name PT with optimistic bonus, value difference and random weighted term (Equation (11)) trajectories as PT (OB + VD + RW).

Additionally, we evaluate the performance of our algorithm under various length of queried trajectories. The experimental results in the Figure 8 and Figure 9 show that the advantage of the random weighted trajectories, especially under the longer queried trajectories.

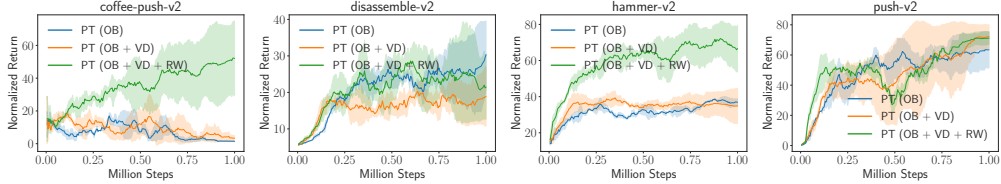

Figure 8: Modular ablation study on the Meta-World tasks with queried trajectory length 100. Tasks from left to right are `coffee-push-v2,disassemble-v2,hammer-v2,` and `push-v2`.

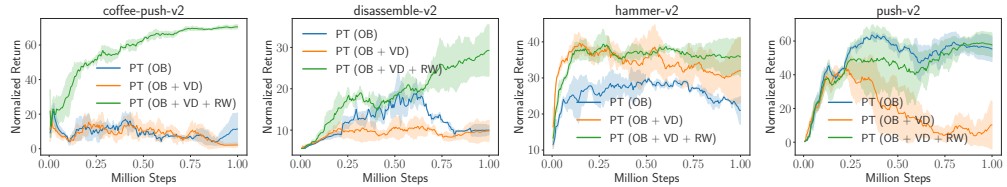

Figure 9: Modular ablation study on the Meta-World tasks with queried trajectory length 200. Tasks from left to right are `coffee-push-v2,disassemble-v2,hammer-v2,` and `push-v2`.

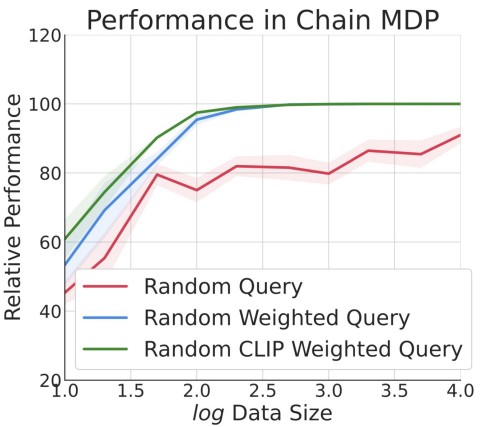
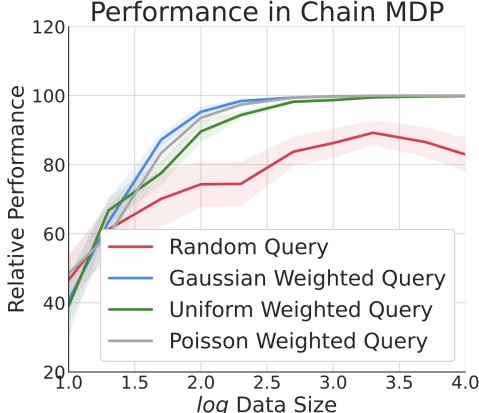

(a) Performance comparison in the chain MDP between radnom weight and clip random weight

(b) Performance comparison in the chain MDP between various random weight.

Figure 10: Ablation studies on the Chain MDP.

# H   EXPERIMENT DETAILS

**OPRL:**   We use the official implementation [1], which uses 7 ensembles. Each ensemble is initially trained with 1 randomly selected query and then performs 3 rounds of active querying and training, and in each round, 1 query is acquired, making a total of 10 queries.

**PT:**   We use the official implementation [2]. We follow its original hyper-parameter settings, and change the number of queries to 10.

**OPRIDE:**   Our code is built on PT. We use the same transformer architecture and hyper-parameter with PT. The ensemble number $N$ is 5. The hyper-parameter $k$ in optimistic bonus is 0.1. The size of $\mathcal{D}_{\text{off}}^{\text{aug}}$ is 10000. The offline pre-training step for $\widetilde{V}_i(\cdot, \cdot)$ in the Equation 13 is $10000 \times c$, where $c$ is the $c$-th selected query. Please refer to Table 4 for detailed parameters.

| Hyperparameter | Value |
| --- | --- |
| Optimizer | Adam |
| Critic learning rate | 3e-4 |
| Actor learning rate | 3e-4 |
| Mini-batch size | 256 |
| Discount factor | 0.99 |
| Target update rate | 5e-3 |
| IQL parameter $\tau$ | 0.7 |
| IQL parameter $\alpha$ | 3.0 |
| Query Number | 10 |
| **OPRL** | **Value** |
| Ensemble Number | 7 |
| **OPRIDE** | **Value** |
| Ensemble Number $N$ | 5 |
| Optimistic Bonus $k$ | 0.1 |
| Size of $\mathcal{D}_{\text{off}}^{\text{aug}}$ | 10000 |
| Offline Pre-training step | $10000 \times c$ |

Table 4: Hyper-parameters sheet of Algorithms.

---

[1] https://github.com/danielshin1/oprl
[2] https://github.com/csmile-1006/PreferenceTransformer/tree/main

