# OpenReview forum: "Query-Efficient Offline Preference-Based Reinforcement Learning via In-Dataset Exploration"
_ICLR.cc/2024/Conference — Submitted to ICLR 2024_

### Official Review · Reviewer_HCQm · 2023-10-30

**Soundness:** 3 good
**Presentation:** 3 good
**Contribution:** 3 good
**Rating:** 3
**Confidence:** 3

**Summary:**

This paper studies improving preference-based RL in offline settings via in-dataset exploration. They proposed an algorithm, called OPRIDE. First, they show that this algorithm achieves statistical efficiency by establishing a bounded suboptimality. Then, they developed a practical version of this algorithm, and experiments show this approach outperforms other methods.

**Strengths:**

This paper is easy to follow. The description of the algorithm is clear, and the theoretical claims are also clearly stated.

The theoretical part looks sound to me, although I haven't checked the proof in detail.

**Weaknesses:**

This paper has 10 pages. I am not sure if it violates the paper length limit. Regarding its content, I have the following concerns.

I don't generally understand what "query-efficient" means in this paper, especially in the theoretical part (section 3.1). If I understand it correctly, the authors proposed an algorithm and showed that it achieves good suboptimality upper bound. Then how is small optimality related to "query efficiency"? I would expect "query efficiency" to have something to do with active learning. Otherwise it is simply "statistical efficiency".

Another concern is that there seems to be a big gap between the theoretical algorithm and the practical one (i.e. algorithm 1 vs algorithm 2). All theoretical results are developed for alg 2 while the experiments are conducted for alg 1. I think this is acceptable only when the gap between the two algorithms is small. However, I feel that the gap is large in my current understanding. For instance, alg 2 assumes a query oracle but alg 2 doesn't. Alg 1 uses an optimistic bonus, but no correspondence is found for alg 1. In addition, Alg 1 is heuristic and lacks some explanation. For example, why the loss functions in (14) and (15) are introduced is not explained (eg, why using expectile loss). The intuitive explanation of adding clipped Gaussian noise is not provided, which may be confusing.

**Questions:**

I didn't understand Tables 1 and 2. I am not sure what these numbers in the tables are. I would suppose they are the output of the learned reward model. However, this confused me even more since comparing the absolute reward value seems meaningless under the Bradley-Terry model (because the distribution will not change under reward shift).

The authors claim to use 10 queries in the experiments. I am not sure how complicated the experiment is, but 10 sounds really small. Could the authors explain more about it?

---

> ### Author Response · Authors · 2023-11-22
> **Response to Reviewer HCQm (Part 1/2)**
>
> Dear Reviewer,
>
> We thank the Reviewer for the constructive comments and clarify your concerns as follows.
> We would appreciate it if you have any further
> feedback.
>
> **W1: Does this paper violate the paper length limit?**
>
> **A for W1:**
> This paper does not violate the paper length limit.
> According to the ICLR guidelines, the reproducibility statement does not count toward the page limit.
>
> Please refer to the author guideline (https://iclr.cc/Conferences/2024/AuthorGuide) for more details.
>
> **W2: How is a small suboptimality related to "query efficiency"?**
>
> **A for W2:**
> A smaller suboptimality bound indicates that we need fewer queries to achieve the same performance bound and, hence, better statistical efficiency. We use "query efficiency" to refer to the fact that our method aims to reduce the number of required preference queries. At each iteration, our method uses an online exploration objective to select a query to ask for preference labels, which can be seen as a special form of active learning.
>
>
> **W3.1: Gap between Algorithm 1 and Algorithm 2.**
>
> **A for W3.1:**
> We respectfully disagree with this assessment. Algorithm 1 is an extension of Algorithm 2 in the deep learning setting without significant gaps. Specifically,
>
> 1. **Query Oracle.** Both Algorithms 1 and 2 require querying the preference between two given trajectories, as standard in preference-based RL. There is no difference in the requirement for query oracles between Algorithms 1 and 2.
>
> 2. **Optimistic Bonus.** The optimistic bonus is used for selecting policy candidates that are possibly optimal, which aligns with the theoretical formulation for the near-optimal policy set as in Equation (8). In fact, Equation (8) can be rewritten as the policy set where every policy is an optimal policy under some optimistic Q-functions. Then, Algorithm 2 can be rewritten using an optimistic bonus, as in some prior works [2]. The variance-based bonus in Equation (12) reflects the uncertainty over the reward functions, which aligns well with the confidence set formulation as in Equation (7).
>
> 3. **Expectile Loss.** The expectile loss in Equations (14) and (15) is used as a conservative estimation for the value function, as proposed in IQL [1]. This aligns well with the use of PEVI in Algorithm 2, which uses a pessimistic value estimation to exploit the offline dataset.

---

> ### Author Response · Authors · 2023-11-22
> **Response to Reviewer HCQm (Part 2/2)**
>
> **W3.2 The intuitive explanation of adding clipped Gaussian noise.**
>
> **A for W3.2:**
> The motivation to add weights is to learn better credit assignments with trajectory-wise information.
> Since we are only allowed to query the preference between trajectories, it is possible that
> we can not learn proper temporal credit assignment with only fixed data.
>
> For example, suppose we have two trajectories $\tau\_1$ and $\tau\_2$ that finish a task with two steps, knowing the preference between $\tau\_1$ and $\tau\_2$ give us no knowledge about how well they do at each step.
> Since we are not allowed to generate new trajectories, a natural idea is to give weights to different parts of the trajectory and ask for the preference again.
>
> Specifically, we can give high weights for step 1 and low weights for step 2 to distinguish which trajectory performs better at step 1.
> Therefore, augmenting trajectories can help for better temporal credit assignment and efficient preference learning.
>
> We have added experiments on Meta-world tasks to further demonstrate the advantage of using weighted trajectories for queries, as shown in Appendix G. We have made this point more clear in the revised version.
>
> As to the choice of using clipped Gaussian noise, we clip the range of the weight to avoid large weights that can lead to high variance in the learning process, as commonly used in practice, like in TD3.
> We also demonstrate the importance of weight clipping in the chain MDP experiment shown in Figure 10 (a) in Appendix G. We have made this motivation more clear in the revised version.
>
> **Q1: What these numbers in the table 1 and 2 are?**
>
> **A for Q1:**
> The numbers in Tables 1 and 2 represent the (normalized) performance of the offline RL algorithm using the reward labels learned via the given preference-based reward learning algorithm. A higher performance indicates a better learned reward function.
>
> **Q2: Why 10 queries work well?**
>
> **A for Q2:**
> Theoretically, preference queries contain rich information such that learning from preferences is not harder than standard RL [3].
> Our theoretical result shows that offline PbRL can be made more efficient in the number of queries by leveraging the knowledge from the offline dataset. A similar phenomenon is also observed in prior and concurrent works [4,5].
>
> Another possible reason is that offline algorithms are known to be robust to the reward functions [6,7,8]. Therefore, we can have a good performance as long as the reward function learned from preference queries does not deviate too much from the true reward functions.
>
>
> Thanks again for the valuable comments.
> We hope our additional experimental results and explanation have cleared your concerns, and we sincerely hope the reviewer can re-evaluate our paper based on our response.
> More comments on further improving the presentation are also very much welcomed.
>
> [1] Kostrikov, Ilya, Ashvin Nair, and Sergey Levine. "Offline reinforcement learning with implicit q-learning." arXiv preprint arXiv:2110.06169 (2021).
>
> [2] Jin, Chi, et al. "Provably efficient reinforcement learning with linear function approximation." Conference on Learning Theory. PMLR, 2020.
>
> [3] Wang, Yuanhao, Qinghua Liu, and Chi Jin. "Is RLHF More Difficult than Standard RL? A Theoretical Perspective." Thirty-seventh Conference on Neural Information Processing Systems. 2023.
>
> [4] Shin, Daniel, Anca D. Dragan, and Daniel S. Brown. "Benchmarks and algorithms for offline preference-based reward learning." arXiv preprint arXiv:2301.01392 (2023).
>
> [5] Anonymous. "Flow to Better: Offline Preference-based Reinforcement Learning via Preferred Trajectory Generation." https://openreview.net/forum?id=EG68RSznLT
>
> [6] Shin, Daniel, Anca D. Dragan, and Daniel S. Brown. "Benchmarks and algorithms for offline preference-based reward learning." arXiv preprint arXiv:2301.01392 (2023).
>
> [7] Li, Anqi, et al. "Survival Instinct in Offline Reinforcement Learning." arXiv preprint arXiv:2306.03286 (2023).
>
> [8] Hu, Hao et al. “Unsupervised Behavior Extraction via Random Intent Priors.” arXiv preprint arXiv:2310.18687 (2023).

---

### Official Review · Reviewer_8oYp · 2023-10-31

**Soundness:** 3 good
**Presentation:** 3 good
**Contribution:** 2 fair
**Rating:** 8
**Confidence:** 3

**Summary:**

This paper presents OPRIDE, an offline preference-based reinforcement learning (PbRL) algorithm that selects trajectories to query based on a pessimistic-value, optimistic-reward approach. Additionally, OPRIDE uses random weights to augment the offline dataset.

Moreover, this paper presents a mathematical justification for using its pessimistic-value, optimistic-reward approach, setting a likely bound on the suboptimality of the policy value estimation.

Experiments in AntMaze and MetaWorld show OPRIDE beating recent baselines such as Preference Transformer. For instance, across 30 MetaWorld tasks and 5 runs, OPRIDE increases the average reward by 27%.

**Strengths:**

* As far as I could follow, mathematically robust motivation, with a bound on the suboptimality of the value estimation.
* The evaluation results are impressive, beating Preference Transformer in the MetaWorld tasks.

**Weaknesses:**

* _W1_: Though the individual sections are really well written, I had trouble following the paper. To mitigate this, I would suggest better delineating the article in the introduction, highlighting key take-aways in each section, adding a summary of Pessimistic Value Iteration (which OPRIDE builds on) to the main paper, and providing more examples. The mathematical analysis is very general (this is a strength of the paper), but providing examples of how it is instantiated for linear MDP with Bradley-Terry preferences would aid in comprehension.
* _W2_: There is still a question whether the same gains observed in MetaWorld and AntMaze apply to other environments such as mujoco-gym and perhaps from human preferences too. (Preference Transformer tested against all these environments)

**[[Post-rebuttal update]]**

The authors revised the manuscript, adding more "sign-posting" and repeating the motivations for each section. As a result the paper is easier to read, thus addressing _W1_. The manuscript remains rather mathematical, but that is the nature of the research.

As for _W2_, the authors conducted additional experiments with mujoco-gym, but OPRIDE surpassing or equalling Preference Transformer (albeit by a lower margin than in Metaworld).

**Questions:**

* _Q1_: In definition 2, the instantiation of $\phi$ for linear MDP with the Bradley-Terry preference model depends on another $\phi$. Is this a mistake? Or is the definition truly recursive?
* _Q2_: What is $T$ on the theoretical guarantees section? Do you mean $K$ the number of queries?
* _Q3_: In figure 4, for `push-v2`, both baselines beat OPRIDE at K>15. Why is that? Is there a downside to OPRIDE as the number of queries goes up?

**[[Post-rebuttal update]]**

The authors addressed all questions and updated the manuscript accordingly. See discussion for more details.

**Nitpicks and suggestions**

* When introducing equation 5, explain that SubOpt is the sub-optimality measure, it will help readers later on.
* In equation 6, mention that $o$ is the ground-truth human preference.
* In section 3.2 $N$ is meant to use the number of reward functions, but $N$ was earlier used to indicate the size of the offline dataset.
* In "Answer to Question 1", _tables 3 and 2_, should be _tables 1 and 2_ instead.

---

> ### Author Response · Authors · 2023-11-22
> **Response to Reviewer 8oYp**
>
> Dear Reviewer,
>
> Thanks for your positive and insightful comments. We provide clarification to your concerns below.
> We would appreciate it if you have any further feedback.
>
> **W1: More clear presentation.**
>
> **A for W1:**
> We thank the Reviewer for pointing this out. As suggested, we highlighted key takeaways in each section, added a summary of Pessimistic Value Iteration to the main paper, and provided more examples in the revised version.
>
> **W2: Whether the same gains observed in MetaWorld and Antmaze apply to other environments such as mujoco-gym and perhaps from human preferences too.**
>
> **A for W2:**
>
> To investigate whether our method is generally applicable to other environments, we conduct experiments on three mujoco tasks, including hopper, walker2d, and halfcheetah. As shown in Table 1, our method outperforms the baseline method PT, especially on the hopper tasks.
>
>
> | | OPRIDE | PT |
> | :---: | :---: | :---: |
> | hopper-medium-v2 | **47.5$\pm$2.3** | 36.9$\pm$2.1 |
> | hopper-medium-expert-v2 | **72.1$\pm$1.5** | 68.0$\pm$2.6 |
> | walker2d-medium-v2 | 71.8$\pm$2.4 | 71.7$\pm$2.6 |
> | walker2d-medium-expert-v2 | 108.5$\pm$0.2 | 109.4$\pm$0.3 |
> | halfcheetah-medium-v2 | 42.4$\pm$0.2 | 42.1$\pm$0.1 |
> | halfcheetah-medium-expert-v2 | **88.8$\pm$0.8** | 81.9$\pm$0.1 |
> | Average | **71.8** | 68.3 |
>
> Table 1. Additional experimental results on Mujoco tasks.
>
>
> **Q1: In definition 2, does the instantiation of linear MDP with the Bradley-Terry preference model depend on another $\phi$? Is the definition truly recursive?**
>
> **A for Q1:**
> A linear MDP with the Bradley-Terry preference model does not depend on another $\phi$. Definition 2 is only used to show that a linear MDP with the Bradley-Terry preference model naturally satisfies the definition of a generalized linear preference model. In linear MDP, the reward function is linear with respect to the feature map $\phi(s,a)$, i.e., $r(s,a)=\theta^T \phi(s,a)$, and the preference over trajectories $\tau\_1$ and $\tau\_2$ can be written as
>
> $$ P(\tau\_1<\tau\_2) = \sigma ( \sum\_{s,a \in \tau\_1} \theta^T\phi(s,a) -  \sum\_{s,a \in \tau\_2} \theta^T\phi(s,a)) $$
> $$ = \sigma ( \theta^T(\sum\_{s,a \in \tau\_1} \phi(s,a) -  \sum\_{s,a \in \tau\_2} \phi(s,a))) $$
>
> Therefore, by overloading the notation $\phi$ and let $\phi(\tau\_1,\tau\_2) = \sum\_{s,a \in \tau\_1} \phi(s,a) -  \sum\_{s,a \in \tau\_2} \phi(s,a)$, a linear MDP with the Bradley-Terry preference model naturally satisfy the definition of generalized linear preference model.
>
> **Q2: What is $T$ on the theoretical guarantees section? and Does $K$ mean the number of queries?**
>
> **A for Q2:**
> Thanks for pointing this out. This is a typo, and it should be $K$ rather than $T$ in the theoretical guarantee section. Yes, $K$ means the number of queries. We have corrected this in the revised version.
>
> **Q3: Why do both baselines beat OPRIDE at $K>15$ for push-v2 in Figure 4? And is there a downside to OPRIDE as the number of queries goes up?**
>
> **A for Q3:**
> This is because push-v2 is a relatively simple task, and the performance saturates with a few queries. To verify this,
> We evaluate the performance of the algorithms with 30 queries on push-v2, and the result is shown in Figure 4. We can see that OPRIDE performs well compared with other methods when the number of queries goes up.
>
> **S: Nitpicks and suggestions.**
>
> **A for S:**
> We appreciate the valuable suggestions of the reviewer, and we have corrected them in the revised version.
>
> Thanks again for the valuable comments.
> We sincerely hope our response has cleared your remaining concerns.

---

> > ### Comment · Reviewer_8oYp · 2023-11-22
> > **Response to rebuttal**
> >
> > Thank you for detailed rebuttal. In what follows I will answer to each of the points raised.
> >
> > --------
> >
> > **W1** Thank you for the revisions, I have revised the additions to the manuscript  and they indeed aid in comprehension.
> >
> >
> > **W2** Thank you for the additional experiments. It is reassuring to see that OPRIDE outperform Preference Transfomer in these tasks.  However, for walker-walk (and to an extent for halfcheetah) the gains are relatively minor. Do authors have an hypothesis as to why? Is it that the tasks are relatively simple? Or is there any other insight we can extract for OPRIDE?
> >
> > **Q1** Thank you for the clarification. This makes sense to me. I realise this is a minor point of the paper, but I suggest authors add the intermediate step $\phi(\tau_1, \tau_2) = \sum_{s,a \in \tau_1} \phi(s,a) - \sum_{s,a \in \tau_2} \phi(s,a)$ to the text.
> >
> > **Q2** Thank you for the clarification.
> >
> >
> > **Q3** Thanks for updating the figure, the explanation makes sense. I also appreciated the addition of standard deviations to the figure.
> >
> >
> > -----
> > I am aware that we are very close to the deadline (and that depending on your locale you may preparing for a holiday), but if at all possible I would be very thankful for an answer to the follow-up question in **W2**.

---

> ### Author Response · Authors · 2023-11-22
>
> Dear Reviewer,
>
> Thanks again for your valuable comments. We provide a response below for the points you raised.
>
> **W2 The improvement on some Mujoco tasks is relatively minor.**
>
> **A for W2**
> This is because Mujoco locomotion tasks have a simple reward function such that a simple preference-based reward learning method can lead to a saturated performance. To verify this, we calculate the correlation coefficient between each dimension of the observation and the reward. As shown in Table 1, the observation can have a very high correlation (>0.9) with the reward in some dimension for every locomotion task, making the reward learning easy on these tasks. This phenomenon is also observed in concurrent works like [1]. Please refer to https://anonymous.4open.science/r/ICLR2024-7244-4A6F/README.md for a visualization of the correlation.
> Therefore, simple methods can also lead to reasonable performance on these tasks, and we believe that tasks like Meta-world, where learning the reward function is more challenging, are a better test bed for query efficiency.
>
> | Tasks |  |  |  |  |  |  |  |  |  |  |  |
> | :---: | :---: | :---: | :---: | :---: | :---: | :---: | :---: | :---: | :---: | :---: | :---: |
> | Hopper | 0.1 | 0.44 | 0.11 | 0.11 | 0.66 | **0.99** | 0.15 | 0.47 | 0.10 | 0.30 | 0.02|
> | Walker2d | 0.18 | 0.40 | 0.20 | 0.32 | 0.00 | 0.27 | 0.18 | 0.04 | **0.97** | 0.46 | 0.63 |
> | | 0.37 | 0.34 | 0.06 | 0.37 | 0.30 | 0.02|  |  |  |  |  |
> | Halfcheetah | 0.07 | 0.10 | 0.15 | 0.05 | 0.08 | 0.06 | 0.07 | 0.01 | **0.91** | 0.11 | 0.10 |
> | | 0.12 | 0.03 | 0.13 | 0.01 | 0.03 | 0.14 |  |  |  |  |  |
>
> Table 1. The correlation coefficient between each dimension of observation and the reward function.
>
>  **Q1 Add the intermediate step to text**
>
>  **A for Q1**
>  We thank the Reviewer for the suggestion, and we have added it to the manuscript.
>
>
> We sincerely hope that our response can address your remaining concerns. More suggestions and discussions are welcomed.
>
> [1] Anonymous. Flow to Better: Offline Preference-based Reinforcement Learning via Preferred Trajectory Generation. https://openreview.net/forum?id=EG68RSznLT

---

> > ### Comment · Reviewer_8oYp · 2023-11-22
> >
> > Thank you for your swift response, I appreciate it. The explanation for the performance difference makes sense to me, and I recommend adding it as context to the appendix with the Mujoco results.
> >
> > Based on the fact that all the weaknesses and questions I have raised have been addressed, I have increased my review rating.

---

> > > ### Author Response · Authors · 2023-11-22
> > > **Thanks for raising the score to 8!**
> > >
> > > We would like to thank the reviewer for raising the score!
> > > We really appreciate the valuable comments and suggestions from the reviewer.

---

### Official Review · Reviewer_i96V · 2023-10-31

**Soundness:** 3 good
**Presentation:** 2 fair
**Contribution:** 2 fair
**Rating:** 5
**Confidence:** 4

**Summary:**

The authors propose an algorithm for query-efficient offline preference-based RL. In this setting, the dataset of trajectories is fixed and the agent must choose pairs of to submit for a preference.

First, they assume a linear transition and reward model. Building on top of the PEVI method, they construct a confidence set of reward functions using a projected MLE estimator. Given that confidence set, they construct a set of near-optimal policies. In order to explore maximally, they choose the pair of policies for which there are two reward functions in the confidence set under which these policies achieve maximally different values. They show that the suboptimality of the method is bounded by the sum of two terms: one that is O(1/sqrt(N)) in the dataset size and one that is O(1/sqrt(K)) in the number of comparisons.

Next, they give an implementable algorithm that works with a static offline dataset. It seems that they augment each trajectory L times with L sampled clipped Gaussians. They train N reward functions by a preference transformer method and compute an optimistic reward function using ensemble disagreement. At each round, the agent chooses the trajectories for which the weighted sum of reward predictions has the largest difference between 2 of the ensemble members. Finally, using IQL, they train optimistic V and Q functions and extract a policy.

They evaluate the implementation on a set of Meta-World manipulation tasks as well as Antmaze. In each they make 10 pairwise queries prior to evaluation the policy performance. They compare against  a handful of recent methods for offline preference-based reward learning. In the experiments, it is clear that this method performs best on the set of manipulation + Antmaze benchmarks. Afterwards, the authors proform studies of the number of queries required to achieve good performance and an ablation where they remove the optimism and the selection method and see how that affects performance.

**Strengths:**

A substantive assessment of the strengths of the paper, touching on each of the following dimensions: originality, quality, clarity, and significance. We encourage reviewers to be broad in their definitions of originality and significance. For example, originality may arise from a new definition or problem formulation, creative combinations of existing ideas, application to a new domain, or removing limitations from prior results. You can incorporate Markdown and Latex into your review. See https://openreview.net/faq.

The problem setting is reasonable and seems like a good way of supervising given large offline dynamics datasets and a hard-to-compute reward function. The theoretical setup seems reasonable and in line with prior work, and it seems like the bounds are interpretable. The theoretical algorithm seems broadly of the right flavor to solve a problem like this.

Broadly speaking, the experimental section shows good results and some ablations and further empirical study is conducted. I don’t think there are yet many methods that address this problem and there is therefore not a wide list of methods to compare against.

**Weaknesses:**

I think the exposition and presentation of this paper is poor. I walked away from the first read without a new insight into the nature of preference based learning. I’m sure there is one to be had in this paper but the lack of writing to inform rather than to describe procedures makes it far more difficult to take one away. In particular:
- The motivation for the augmentation with a truncated Gaussian missed me entirely.
- The chain experiment is not described well in the text or the appendix.
- The justification for the use of projected MLE or even the definition of the projected MLE is missing from the method section.
- There are a substantial number of typos and errors in the writing.
- It's not clear to me what the counterfactual trajectories are.
See questions for things I concretely would like to know in order to increase my score.

**Questions:**

At the end of 3.1 you mention that “querying with an offline dataset can be much more sample efficient when N >> T”. What is T here?
What is “True Reward” in table 1 / 2?
Can you explain why you augment the trajectories with a truncated Gaussian?
Why a truncated Gaussian rather than some other random variable?
Are we sure that an ensemble of reward functions is going to give reasonable uncertainty estimates of the preference model?  What if preferences have high aleatoric uncertainty?
How is the optimization problem in (13) solved?
What do counterfactual trajectories have to do with weighting?
How does this policy error bound compare to e.g. a G-optimal design and then PEVI? The Pacchiano et al result?
What are the steps in the x axis of Figure 3?

---

> ### Author Response · Authors · 2023-11-22
> **Response to Reviewer i96V (Part 1/2)**
>
> Dear Reviewer,
>
> Thanks for your valuable comments. We have provided additional experimental results and explanations to address your concerns, and we hope the following clarifications shed light on the points you raised.
>
> **W1: The motivation for the augmentation.**
>
> **A for W1**
> The motivation to add weights is to learn a better credit assignment with trajectory-wise information.
> Since we are only allowed to query the preference between trajectories, it is possible that
> we can not learn proper temporal credit assignment with only fixed data.
>
> For example, suppose we have two trajectories $\tau\_1$ and $\tau\_2$ that finish a task with two steps, knowing the preference between $\tau\_1$ and $\tau\_2$ give us no knowledge about how well they do at each step.
> Since we cannot generate new trajectories, a natural idea is to give weights to different parts of the trajectory and ask for the preference again.
>
> Specifically, we can give high weights for step 1 and low weights for step 2 to distinguish which trajectory performs better at step 1.
> Therefore, augmenting trajectories can help for better temporal credit assignment and efficient preference learning.
> We have added experiments on Meta-world tasks to demonstrate further the advantage of using weighted trajectories for queries, as shown in Appendix G. We have made this point more clear in the revised version.
>
> **W2: Details of the chain MDP experiment.**
>
> **A for W2:**
> We thank the reviewer for pointing this out, and we gave a more detailed description of the chain MDP experiment in Appendix E in the revised version.
>
> **W3: Justification for the use of projected MLE and  the definition of the projected MLE.**
>
> **A for W3:**
> By assumption, the parameter $\theta$ for the reward function is bounded, but an MLE estimator can generally be unbounded. Therefore, we project the MLE estimator to the space with bounded norms, ensuring the estimation is in the confidence set [1,2]. We have made this motivation more clear in the revised version.
>
> **W4: Typo errors.**
>
> **A for W4:**
> We thank the reviewer for pointing it out, and we have substantially revised the manuscript.
>
> **W5 & Q8: What the counterfactual trajectories are? What do counterfactual trajectories have to do with weighting?**
>
> **A for W5  Q8:**
> The counterfactual trajectories in Figure 2(c) refer to the two colored trajectories that are out of the dataset. We can not infer the preference between these two out-of-dataset trajectories if weighted queries are not allowed.
>
> As explained in W1, by adding weights to trajectories, we can learn better credit assignments.
> We can use the former 2-stage task in W1 as an example.
> Without weighted queries, we can not infer the preference between the counterfactual trajectories $\tau'\_1$ and $\tau'\_2$, where $\tau'\_1$ acts as $\tau\_1$ at step 1 and acts $\tau\_2$ at step 2, and $\tau'\_2$ acts as $\tau\_2$ at step 1 and acts $\tau\_1$ at step 2.
> With weighted queries, we can identify the preference at each step.
> Therefore, weighted queries help reason about preferences between
>  the counterfactual trajectories $\tau'\_1$ and $\tau'\_2$ without generating new trajectories.
>
> **Q1: What is $T$ in “querying with an offline dataset can be much more sample efficient when N >> T” at the end of Section 3.1?**
>
> **A for Q1:**
> Thanks for pointing it out. This is a typo, and it should be $K$, the number of comparisons. Intuitively, online preference-based RL requires online samples to learn both dynamics and the preference function. In contrast, in offline settings, the dynamics information is already contained in the dataset, and we only need to infer the preference function, which requires fewer samples than the online one.
>
> **Q2: What is ``True Reward'' in the table 1/2?**
>
> **A for Q2:**
> In Table 1/2, ``True Reward'' denotes the performance of offline RL algorithms under the original reward function of the dataset, which can be seen as the oracle performance for preference-based reward learning methods.
> We have added a detailed explanation in the revised version to enhance clarity.
>
> **Q3 & Q4: Can you explain why you augment the trajectories with a truncated Gaussian? Why a truncated Gaussian rather than some other random variable?**
>
> **A for Q3 & Q4:**
> The motivation for using weighted queries is explained in W1.
>
> We clip the range of the weight to avoid large weights that can lead to high variance in the learning process, as commonly used in practice [3].
> We conduct additional ablation studies to demonstrate the importance of weight clipping in the chain MDP experiment shown in Figure 10 (a) in Appendix G.
>
> Using other random variables is also possible, but we found that Gaussian weights perform better in practice. Please refer to Appendix G's Figure 10 (b) for an ablation study.

---

> ### Author Response · Authors · 2023-11-22
> **Response to Reviewer i96V (Part 2/2)**
>
> **Q5 & Q6: Is an ensemble of reward functions going to give reasonable uncertainty estimates of the preference model? What if preferences have high aleatoric uncertainty?**
>
> **A for Q5 & Q6:**
> It is a common practice to use an ensemble of reward functions to estimate the uncertainty of the preference model, and they are known to lead to good empirical performance [4,5]. Under the Bradley-Terry preference model, the aleatoric uncertainty is accounted for as a soft probability rather than a one-hot probability. In general cases, advanced methods like bootstrapping have the potential to improve uncertainty estimation further and can separate aleatoric and epistemic uncertainties [6], which is an interesting direction for future work.
>
>
> **Q7: How is the optimization problem in (13) solved?**
>
> **A for Q7:**
> Equation (13) takes the pair of trajectories that maximize the value difference. Note that there is no optimization procedure (for continuous variables) needed, and we only need to search for the best candidate, which requires $O(M^2N^2)$ time, where $M$ is the number of sampled trajectories, and $N$ is the number of Q-values.
>
>
> **Q9: How does this policy error bound compare to, e.g., a G-optimal design and then PEVI?**
>
> **A for Q9:**
> Without weighted queries, pure offline methods are known to suffer from an exponential policy error bound [7]. When using weighted queries, a G-optimal design and then PEVI would yield a similar performance bound, but the second term consisting of the number of queries $K$ in Equation (10) would depend on an additional coverage coefficient [8]. This can lead to suboptimal performance when the dataset has poor coverage, which motivates us to introduce exploration into offline PbRL.
>
> **Q10: How does this policy error bound compare to the Pacchiano et al result?**
>
> **A for Q10:**
> Pacchiano et al. focus on the pure online setting, considering two cases: a known model and an unknown one to be learned online. Our result considers the offline case, where the model is estimated via an offline dataset, which requires a careful treatment balancing offline exploitation (for dynamics estimation) and online exploration (for preference learning).
>
>
> **Q11: What are the steps in the $x$ axis of Figure 3?**
>
> **A for Q11:**
> Figure 3 depicts the performance of the offline RL algorithm as a variable of the offline training procedure.
> The $x$-axis represents the training steps of the offline RL algorithm. We have made this clear in the revised version.
>
>
> Thanks again for the valuable comments.
> We sincerely hope our additional
> explanations have cleared the concern.
> More comments on further improving the presentation are also very much welcomed.
>
> [1] Faury, Louis, et al. "Improved optimistic algorithms for logistic bandits." International Conference on Machine Learning. PMLR, 2020.
>
> [2] Filippi, Sarah, et al. "Parametric bandits: The generalized linear case." Advances in neural information processing systems 23 (2010).
>
> [3] Fujimoto, Scott, Herke Hoof, and David Meger. "Addressing function approximation error in actor-critic methods." International conference on machine learning. PMLR, 2018.
>
> [4] Christiano, Paul F., et al. "Deep reinforcement learning from human preferences." Advances in neural information processing systems 30 (2017).
>
> [5] Shin, Daniel, Anca Dragan, and Daniel S. Brown. "Benchmarks and Algorithms for Offline Preference-Based Reward Learning." Transactions on Machine Learning Research (2022).
>
> [6] Chua, Kurtland, et al. "Deep reinforcement learning in a handful of trials using probabilistic dynamics models." Advances in neural information processing systems 31 (2018).
>
> [7] Zhan, Wenhao, et al. "Provable Offline Reinforcement Learning with Human Feedback." arXiv preprint arXiv:2305.14816 (2023).
>
> [8] Agarwal, Alekh, et al. "Reinforcement learning: Theory and algorithms." CS Dept., UW Seattle, Seattle, WA, USA, Tech. Rep 32 (2019).

---

### Author Response · Authors · 2023-11-22
**Rebuttal Summary**

Dear Reviewers,

We thank all the reviewers for their constructive and valuable comments. We are encouraged to learn that many found our work "empirically impressive," "theoretically sound," and have a "clear description." We genuinely appreciate these positive remarks.

We also acknowledge the concerns raised by the reviewers. We have revised our manuscript for more details and better presentation and added several experiments as a response. Specifically,

1. We added more explanations in the introduction and method section to motivate and present our method better.

2. We added detailed explanations for the PEVI algorithm, the projected MLE estimator, and the chain MDP experiment.

3. We added an experiment on the Mujoco tasks to show the general applicability of our method.

4. We added an ablation study on weighted queries to demonstrate the importance of the proposed method better, especially when the queried trajectories have a long horizon.

We sincerely hope these updates and clarifications address the Reviewers' concerns. We welcome further discussion and suggestions for improving the paper.

---

### Meta-Review · Area_Chair_FyXd · 2023-12-06

**Metareview:**

The paper proposes OPRIDE, a new algorithm for offline preference-based RL via in-dataset exploration. The algorithm selects trajectories to query based on pessimistic value estimation and optimistic bonus. The authors theoretically analyzed suboptimality gap. Experiments in AntMaze and MetaWorld show OPRIDE outperforms baselines. After extensive discussion between reviewer-author and reviewer-AC, a consensus of rejection was reached among reviewers during the discussion phase. To summarize, the main pros and cons are:

Pros:
- The problem is important and a good solution could be impactful.
- The results with 10 preference queries are very strong.
- All theoretical results look sound.

Cons:
- The presentation lacks clarity.
- The random weight design is concerning as it sacrifices practical usability.

Overall, I recommend rejecting current version. I also encourage the authors to further improve the paper according to the reviews.

**Justification For Why Not Higher Score:**

During the discussion phase, reviewers reached a consensus on rejecting the paper.

**Justification For Why Not Lower Score:**

N/A

---

### Decision · Program_Chairs · 2024-01-16

Reject